# Sharper Generalization Bounds for Pairwise Learning

**Yunwen Lei**[1,2]    **Antoine Ledent**[2]*    **Marius Kloft**[2]

[1]School of Computer Science, University of Birmingham, Birmingham B15 2TT, United Kingdom
[2]Department of Computer Science, TU Kaiserslautern, Kaiserslautern 67653, Germany
`y.lei@bham.ac.uk`  `ledent@cs.uni-kl.de`  `kloft@cs.uni-kl.de`

## Abstract

Pairwise learning refers to learning tasks with loss functions depending on a pair of training examples, which includes ranking and metric learning as specific examples. Recently, there has been an increasing amount of attention on the generalization analysis of pairwise learning to understand its practical behavior. However, the existing stability analysis provides suboptimal high-probability generalization bounds. In this paper, we provide a refined stability analysis by developing generalization bounds which can be $\sqrt{n}$-times faster than the existing results, where $n$ is the sample size. This implies excess risk bounds of the order $O(n^{-1/2})$ (up to a logarithmic factor) for both regularized risk minimization and stochastic gradient descent. We also introduce a new on-average stability measure to develop optimistic bounds in a low noise setting. We apply our results to ranking and metric learning, and clearly show the advantage of our generalization bounds over the existing analysis.

## 1 Introduction

In modern machine learning, we frequently encounter problems where the performance of a model depends on *pairs* of training instances. As examples consider the following. In ranking problems, our aim is to learn a function that can predict the ordering of examples [13, 44]. In metric learning, which plays a key role in clustering problems [9, 28], we wish to learn an adequate distance metric between instances. In AUC maximization, which is deployed to class-imbalanced learning problems, we aim to find a classifier that maximizes the probability of scoring a positive example higher than a negative one [14]. Further examples include learning with minimum error entropy loss functions [27], multiple kernel learning [31], preference learning [22], and gradient learning [41]. All these so-called *pairwise learning* problems involve a loss function based on pairs of training examples. This is in a sharp contrast to classification and regression, where the loss function depends only on a single instance. Those problems are referred to as *pointwise learning* problems.

In machine learning, we frequently build predictive models by optimizing their empirical behavior on training instances, that is, to achieve a small training error. However, a small training error does not imply that the learnt models will generalize well to test examples. Generalization analysis—which is a central topic in statistical learning theory (SLT) [40]—studies the generalization gap between the training and testing errors. There is a large amount of work on the generalization analysis of learning algorithms, largely based on either algorithmic stability [7, 17], complexity analysis of models [3, 52], PAC-Bayesian analysis [38], or integral operators [49, 53]. Most of this work focuses on pointwise learning, while pairwise learning is far less studied. A difficulty occurring in the generalization analysis of pairwise learning is that the objective function is not a sum of identically and independently distributed (i.i.d.) random variables [1, 9, 13, 30]—a fundamental assumption in SLT.

In this paper, we employ the methodology of algorithmic stability for generalization analysis of pairwise learning. Appealingly, algorithmic stability considers just the one prediction function output by the learner [7], while methods based on uniform convergence, such as the Rademacher complexity [3], bound the difference of training and testing errors for *all* prediction functions. The latter approach generally involves a square-root dependency on the input dimension [2, 18, 54]. For comparison, algorithmic stability enables dimension-independent generalization bounds [20].

While there is preliminary work on the algorithmic stability of metric learning and ranking, the resulting generalization bounds are not satisfactory. The best existing bounds decay at the order of $O(\gamma\sqrt{n})$ [1, 28, 55], where $\gamma$ is the uniform-stability constant of the learning algorithm. In regularized risk minimization (RRM), this results in an excess risk bound of order $O(n^{-\frac{1}{4}})$ at best, where $n$ is the number of training examples.

As a main contribution of this paper, we show an improved bound for this setting of order $O(\gamma \log n)$, which translates into $O\big(\log(n)/\sqrt{n}\big)$ for excess risks of RRM. Remarkably, although the bound improves the previously best known rate achieved through stability analysis by a factor of $\sqrt{n}/\log(n)$, it applies more generally: we remove the standard assumption of a bounded loss function used in the prevalent stability analyses [1, 8, 19, 20, 55]. The loss of some of the most commonly used pairwise learning methods—including rankSVM [29] and MPRank [15]—is unbounded, for which we show, for the first time, a stability analysis. Based on our connection between generalization and stability, we also derive, to the best of our knowledge, the first probabilistic generalization bound for stochastic gradient descent (SGD) in pairwise learning. Our result quantifies how to trade-off optimization and generalization to achieve a refined excess risk bound in this setting.

The above bound holds generally for any confidence level, which is informative to understand the variability of the algorithm and is necessary if the algorithm is used many times [20]. Furthermore, we show a sharper bound, but which holds in expectation and in a realizable case (where zero training error is achievable). Such bounds are called *optimistic* bounds in the literature [50]. For this setting, we show an excess risk bound of order $O(n^{-1})$. For the proof, we introduce a new on-average stability measure for pairwise learning and quantify its implication to generalization.

Finally, we consider applications of our general theory to ranking and metric learning, where we obtain generalization bounds with significantly improved dependence on $n$ as compared to the existing stability analysis. Furthermore, our stability analysis also removes the dependency on the complexity of the hypothesis space and the input dimension in the uniform convergence analysis.

**Structure**. We review related work in Section 2, and give background information in Section 3. We list main results in Section 4 and give applications in Section 5. We conclude the paper in Section 6.

## 2 Related Work

In this section, we summarize the related work on the generalization analysis of pairwise learning, which we categorize according to the employed proof techniques.

In complexity (uniform convergence) analysis, we view generalization gaps between training and testing errors as $U$-statistics of order two. We can then bound the supremum of $U$-statistics over the hypothesis space—the $U$-process [9, 13, 34, 44, 54, 58, 61]. To this end, decoupling techniques have been introduced to represent the objective function as a summation of i.i.d. random variables plus a degenerate U-statistic [13]. This approach can yield meaningful generalization bounds of the order $O(1/\sqrt{n})$ for several pairwise learning problems, including ranking [13, 44] and metric learning [9, 54]. The authors of [13, 44] show fast-rate generalization bounds under stronger capacity assumptions on the hypothesis space and Bernstein-type of assumptions on the relationship between variances and expectations. Fast generalization bounds were established for metric learning [58], which, however requires a boundedness assumption on the output model, a bounded gradient assumption and the learning models to be linear. The complexity approach ignores the interaction between the learning algorithm and the training dataset in the search of the output model. It therefore implies generalization bounds depending on the complexity of the hypothesis space [3] and the input dimension. As indicated in [2, 13, 54], a square-root dependency on the dimension is generally inevitable for the uniform convergence in metric learning and ranking. This means that such bounds can quickly become uninformative in high dimensions [1]. For hypothesis spaces with an unbounded complexity, uniform convergence bounds cannot be applied at all [1]. The stability analysis is preferable in both cases.

An advantage of uniform convergence approach is that it is able to imply meaningful generalization bounds in a non-convex learning setting [16, 21, 39]. As a comparison, stability analysis requires very small step sizes to enjoy good stability for non-convex problems [25], which inevitably leads to very slow convergence rates of optimization errors.

The second popular approach studies pairwise learning using algorithmic stability, which is a fundamental concept in SLT dating back to 1970s [46]. The modern framework of stability analysis was established in the seminal paper [7], where an important concept called the *uniform stability* was introduced. This stability measure was then extended to study randomized algorithms [17], to investigate the concentration of output models [35], and to exploit the summation structure of the empirical risk [47]. These stability measures have found applications in privacy learning [4], stochastic optimization [5, 10, 25, 32, 33], and structured prediction [36]. The fundamental role of algorithmic stability in SLT was illustrated by establishing its close connection to learnability [42, 47]. Very recently, elegant high-probability bounds were established for uniformly stable algorithms [8, 19, 20, 37]. The above mentioned stability analysis was conducted in the setting of pointwise learning. There is also some interesting work on the stability analysis of pairwise learning. For example, the connection between generalization and algorithmic stability was established for ranking [1, 23]. Furthermore, it was shown there that kernel-based ranking algorithms in a regularization setting enjoy uniform stability. Algorithmic stability was further used to yield dimension-independent bounds for regularized metric learning [28, 55]. The stability and its trade-off with optimization errors were studied for a variant of SGD in pairwise learning [48], inspired by the recent work in the pointwise learning setting [11, 25].

We now briefly mention related work on the generalization analysis of pairwise learning using other proof techniques than complexity analysis or algorithmic stability. Algorithmic robustness was estimated for pairwise learning [12], which in turn implies generalization bounds [6]. Convex analysis was applied to study the regret bounds and generalization bounds of online pairwise learning [30, 56]. The tool of integral operators was used to exploit the structure of the specific least squares loss functions, where the learnt model can be written in a closed-form [59].

## 3   Background

### 3.1   Pairwise learning

Assume we are given a training dataset $S = \{z_1, \ldots, z_n\}$ drawn independently from a probability measure $\rho$ defined over a sample space $\mathcal{Z} = \mathcal{X} \times \mathcal{Y}$, where $\mathcal{X} \subset \mathbb{R}^d$ is an input space of dimension $d$ and $\mathcal{Y} \subset \mathbb{R}$ is an output space. Based on $S$, we wish to build a model $h : \mathcal{X} \mapsto \mathcal{Y}$ or $h : \mathcal{X} \times \mathcal{X} \mapsto \mathbb{R}$ that can be used to do prediction when we are given some testing examples. We consider a parametric model where the model $h_{\mathbf{w}}$ can be parameterized by an index $\mathbf{w} \in \mathcal{W}$, where $\mathcal{W} \subseteq \mathbb{R}^{d'}$ is a parameter space of dimension $d'$ ($d'$ is not necessarily equal to $d$, and can be infinite). Unlike pointwise learning, a distinctive property of pairwise learning is that the performance of a model should be measured over pairs of training examples. That is, the behavior of $h_{\mathbf{w}}$ over $z, \tilde{z} \in \mathcal{Z}$ is measured by $\ell(\mathbf{w}; z, \tilde{z})$, where $\ell : \mathcal{W} \times \mathcal{Z} \times \mathcal{Z} \mapsto \mathbb{R}_+$ is a loss function. Then, the empirical behavior of $h_{\mathbf{w}}$ can be quantified by the empirical risk $R_S(\mathbf{w})$ defined by

$$R_S(\mathbf{w}) = \frac{1}{n(n-1)} \sum_{i,j \in [n]: i \neq j} \ell(\mathbf{w}; z_i, z_j), \tag{3.1}$$

where we use the notation $[n] = \{1, \ldots, n\}$. We train a predictive model by applying an algorithm $A$ to $S$, for which some popular choices include empirical risk minimization, regularized/structural risk minimization, (stochastic) gradient descent, etc. An algorithm $A$ can be understood as a mapping from $\mathcal{Z}^n$ to $\mathcal{W}$, with $A(S)$ being the output of $A$ when applied to $S$. Typically, the output model $A(S)$ would enjoy a small empirical risk since we are often fitting training examples. However, this does not necessarily mean that it also enjoys a small population risk $R(\mathbf{w}) = \mathbb{E}_{z,\tilde{z}}[\ell(\mathbf{w}; z, \tilde{z})]$, which quantifies the prediction behavior of $\mathbf{w}$ over testing examples. The generalization gap of a model $\mathbf{w}$ is defined as the difference between the population risk and empirical risk, i.e., $R(\mathbf{w}) - R_S(\mathbf{w})$.

We are particularly interested in RRM, where a regularizer $r : \mathcal{W} \mapsto \mathbb{R}_+$ is added into the data-fitting term $R_S$ to increase the regularity of an algorithm. The resulting algorithm then outputs the model by

$$\mathbf{w}_S = \arg \min_{\mathbf{w} \in \mathcal{W}} \left[ F_S(\mathbf{w}) := \frac{1}{n(n-1)} \sum_{i,j \in [n]: i \neq j} f(\mathbf{w}; z_i, z_j) \right], \quad (3.2)$$

where $f : \mathcal{W} \times \mathcal{Z} \times \mathcal{Z} \mapsto \mathbb{R}_+$ is defined as $f(\mathbf{w}; z, \tilde{z}) = \ell(\mathbf{w}; z, \tilde{z}) + r(\mathbf{w})$. Although the above objective function involves $O(n^2)$ terms in the summand, one can use SGD to achieve sample-size independent convergence rates [45]. Let $\mathbf{w}_1 \in \mathcal{W}$. At the $t$-th iteration, SGD first randomly selects $(i_t, j_t)$ from the uniform distribution over the set $\{(i, j) : i, j \in [n], i \neq j\}$, and updates the model by

$$\mathbf{w}_{t+1} = \mathbf{w}_t - \eta_t \left( \ell'(\mathbf{w}_t; z_{i_t}, z_{j_t}) + r'(\mathbf{w}_t) \right), \quad (3.3)$$

where $\{\eta_t\}_t$ is a step size sequence, and $\ell'(\mathbf{w}_t; z_{i_t}, z_{j_t})$ denotes a subgradient of $\ell(\cdot; z_{i_t}, z_{j_t})$ at $\mathbf{w}_t$.

## 3.2 Algorithmic stability

Algorithmic stability plays an important role in studying the behavior of a learning algorithm. Intuitively, we say an algorithm $A : \mathcal{Z}^n \mapsto \mathcal{W}$ is stable if the output model $A(S)$ is insensitive to perturbations of $S$. There are various notions of stability, including uniform stability, hypothesis stability, error stability and on-average stability [7, 17, 47]. A particularly interesting stability measure is uniform stability, which was introduced in [7] and extended in [1] to pairwise learning.

**Definition 1** (Uniform Stability). We say a deterministic algorithm $A : \mathcal{Z}^n \mapsto \mathcal{W}$ is $\gamma$-uniformly stable if for any training datasets $S, S' \in \mathcal{Z}^n$ that differ by at most a single example we have

$$\sup_{z, \tilde{z} \in \mathcal{Z}} \left| \ell(A(S); z, \tilde{z}) - \ell(A(S'); z, \tilde{z}) \right| \leq \gamma.$$

We will use the above notion of uniform stability to develop high-probability generalization bounds. To construct optimistic bounds, we introduce a novel on-average stability for pairwise learning, which is motivated by the recent work on on-average stability for pointwise learning [24, 32, 33, 47]. The difference is that we consider perturbations of a training dataset by two examples.

**Definition 2** (On-average stability). Let $S = \{z_1, \ldots, z_n\}, S' = \{z'_1, \ldots, z'_n\}$ be independently drawn from $\rho$. For any $i < j$, we denote

$$S_{i,j} = \left\{ z_1, \ldots, z_{i-1}, z'_i, z_{i+1}, \ldots, z_{j-1}, z'_j, z_{j+1}, \ldots, z_n \right\}. \quad (3.4)$$

We say a deterministic algorithm $A$ is $\gamma$-on-average stable if

$$\frac{1}{n(n-1)} \sum_{i,j \in [n]: i \neq j} \mathbb{E}_{S,S'} \left[ \ell\left( A(S_{i,j}); z_i, z_j \right) - \ell\left( A(S); z_i, z_j \right) \right] \leq \gamma.$$

It is clear that on-average stability is weaker than uniform-stability since it involves the expectation over training examples and the average of indices. As a comparison, uniform stability involves a supremum over both training examples and testing examples $z, \tilde{z}$.

## 4 Main Results

In this section, we present our main results on generalization bounds based on stability. We always let $\| \cdot \|$ be a norm induced by an inner product $\langle \cdot, \cdot \rangle$ in a Hilbert space, i.e., $\| \cdot \|^2 = \langle \cdot, \cdot \rangle$. Then its dual norm is itself. We say a function $g : \mathcal{W} \mapsto \mathbb{R}$ is $\sigma$-strongly convex w.r.t. a norm $\| \cdot \|$ if

$$g(\mathbf{w}) - \left( g(\mathbf{w}') + \langle \mathbf{w} - \mathbf{w}', g'(\mathbf{w}') \rangle \right) \geq \sigma \|\mathbf{w} - \mathbf{w}'\|^2/2, \quad \forall \mathbf{w}, \mathbf{w}' \in \mathcal{W}.$$

### 4.1 Generalization by algorithmic stability

Our first result (Theorem 1) to be proved in Appendix A is a high-probability generalization bound for uniformly stable algorithms in pairwise learning, motivated by the recent analysis in pointwise learning [8, 19, 20, 37]. One of the key tools we use in the analysis is a concentration inequality from [8], which considers a summation of $n$ functions of $n$ independent random variables. However,

this concentration inequality does not fit the structure of pairwise learning. A difficulty is the inter-dependency among the $n(n-1)$ terms in the objective function. Our novelty to tackle this difficulty is to introduce a new decomposition to exploit the structure of the $U$-statistic in the pairwise objective function (3.1). Below, $e$ denotes the base of the natural logarithm. For any $\alpha \geq 0$, $\lceil \alpha \rceil$ denotes the least integer no smaller than $\alpha$. For any random variable $Z$, we denote by $\mathbb{E}_Z[\cdot]$ the conditional expectation with respect to (w.r.t.) $Z$.

**Theorem 1.** *Let $A : \mathcal{Z}^n \mapsto \mathcal{W}$ be $\gamma$-uniformly stable and $M > 0$. Suppose $\left|\mathbb{E}_S[\ell(A(S); z, \tilde{z})]\right| \leq M$ for all $z, \tilde{z} \in \mathcal{Z}$. Then for all $\delta \in (0, 1/e)$ the following inequality holds with probability $1 - \delta$*

$$|R_S(A(S)) - R(A(S))| \leq 4\gamma + e\Big(12\sqrt{2}M(n-1)^{-\frac{1}{2}}\sqrt{\log(e/\delta)} + 48\sqrt{6}\gamma\lceil\log_2(n-1)\rceil\log(e/\delta)\Big). \tag{4.1}$$

**Remark 1.** We now compare Theorem 1 with the existing stability analysis. Roughly speaking, Theorem 1 shows that the generalization gap for $\gamma$-uniformly stable algorithms decays as $O(\gamma \log n + n^{-\frac{1}{2}})$ with high probability (we ignore $\log(1/\delta)$ for brevity). Under the same conditions, it was shown for pairwise learning that [1, 15, 28, 55]

$$|R_S(A(S)) - R(A(S))| = O\big(\sqrt{n}\gamma + n^{-\frac{1}{2}}\big). \tag{4.2}$$

It is clear that our result significantly improves (4.2) by replacing their dominant term $\sqrt{n}\gamma$ with $\gamma \log n$. Specifically, if $\gamma = O(n^{-\alpha})$ with $\alpha \in (\frac{1}{2}, 1]$ (actually $\gamma = O(1/(n\sigma))$ if $F_S$ is $\sigma$-strongly convex [7]), then (4.1) becomes $|R_S(A(S)) - R(A(S))| = O(n^{-\frac{1}{2}})$, while (4.2) becomes $|R_S(A(S)) - R(A(S))| = O(n^{\frac{1}{2}-\alpha})$. The existing complexity analysis for pointwise learning suggests $\sigma = O(n^{-\frac{1}{2}})$ to get an optimal bound [51, eq (14)]. In this case, $\gamma = O(n^{-\frac{1}{2}})$ and our stability analysis implies the nice bound $O(n^{-\frac{1}{2}} \log n)$, while (4.2) implies the vacuous bound $O(1)$.

## 4.2 Generalization bounds for regularized risk minimization

We now apply Theorem 1 to establish generalization bounds for pairwise learning with strongly convex objective functions. A preliminary step in the application of Theorem 1 is to control $\mathbb{E}_S[\ell(A(S); z, \tilde{z})]$. To this aim, we establish the following lemma to be proved in Appendix B. Let

$$\mathbf{w}^* = \arg\min_{\mathbf{w}\in\mathcal{W}}\big[R(\mathbf{w}) + r(\mathbf{w})\big], \quad \mathbf{w}_R^* = \arg\min_{\mathbf{w}\in\mathcal{W}}R(\mathbf{w}).$$

**Lemma 2.** *Suppose $F_S$ is $\sigma$-strongly convex w.r.t. a norm $\|\cdot\|$. Define the algorithm $A$ as $A(S) = \arg\min_{\mathbf{w}\in\mathcal{W}} F_S(\mathbf{w})$. If $A$ is $\gamma$-uniformly stable, then $\mathbb{E}[\|A(S) - \mathbf{w}^*\|^2] \leq 8\gamma/\sigma$.*

In the existing analysis, one often uses the $\sigma$-strong convexity of $F_S$ to show $\|A(S)\| = O(1/\sqrt{\sigma})$ [9], which implies a suboptimal bound since the convexity parameter $\sigma$ is often very small in practice, i.e., $\sigma = O(n^{-\alpha})$ for $\alpha \in (0, 1)$ ($\sigma$ is roughly the regularization parameter which should decay in this way [19, 51]). As a comparison, Lemma 2 implies that $\mathbb{E}_S[\|A(S) - \mathbf{w}^*\|] = O(\sqrt{\gamma/\sigma})$, which is significantly smaller than $O(1/\sqrt{\sigma})$ since the uniform stability parameter is often very small [7].

We need the following assumption to derive Theorem 3, whose proof is given in Appendix B.

**Assumption 1.** Let $b, \sigma_0 > 0$. We assume $0 \leq \ell(0; z, \tilde{z}) \leq b$ for all $z, \tilde{z} \in \mathcal{Z}$. We also assume $\mathrm{Var}[\ell(\mathbf{w}^*; Z, \tilde{Z})] < \sigma_0^2$, where $\mathrm{Var}[\ell(\mathbf{w}^*; Z, \tilde{Z})]$ denotes the variance of $\ell(\mathbf{w}^*; Z, \tilde{Z})$.

We use the notation $B \asymp \widetilde{B}$ if there exist constants $c_1, c_2 > 0$ such that $c_1\widetilde{B} < B \leq c_2\widetilde{B}$.

**Theorem 3.** *Let Assumption 1 hold and $L \in \mathbb{R}_+$. Define $A$ as $A(S) = \arg\min_{\mathbf{w}\in\mathcal{W}} F_S(\mathbf{w})$. Suppose $F_S$ is $\sigma$-strongly convex w.r.t. $\|\cdot\|$ for all $S$. Assume*

$$\big|\ell(\mathbf{w}; z, \tilde{z}) - \ell(\mathbf{w}'; z, \tilde{z})\big| \leq L\|\mathbf{w} - \mathbf{w}'\|, \quad \forall z, \tilde{z} \in \mathcal{Z}, \mathbf{w}, \mathbf{w}' \in \mathcal{W}. \tag{4.3}$$

*Then for $\delta \in (0, 1/e)$, with probability $1 - \delta$ the generalization gap $R_S(A(S)) - R(A(S))$ satisfies*

$$|R_S(A(S)) - R(A(S))| = O\Big((n\sigma)^{-1}\log n\log(1/\delta) + \sqrt{n^{-1}\log(1/\delta)}\Big). \tag{4.4}$$

*Furthermore, if $r(\mathbf{w}) = O(\sigma\|\mathbf{w}\|^2)$, $\sigma \asymp n^{-1/2}$, $\sup_{z,z'}\ell(\mathbf{w}_R^*; z, z') = O(\sqrt{n})$ and Assumption 1 holds with $\mathbf{w}^*$ replaced by $\mathbf{w}_R^*$, then with probability at least $1 - \delta$ we have the following bound on excess risk $R(A(S)) - R(\mathbf{w}_R^*)$*

$$R(A(S)) - R(\mathbf{w}_R^*) = O(n^{-\frac{1}{2}}\log n\log(1/\delta)). \tag{4.5}$$

**Remark 2.** We present some comparisons with the existing work. Under similar assumptions and additional boundedness assumptions, the existing stability analysis implies the generalization bound $|R_S(A(S)) - R(A(S))| = O(\sigma^{-1} n^{-\frac{1}{2}})$ for pairwise learning with $\sigma$-strongly convex objective functions [1, 15, 28, 55], which can be $\sqrt{n}$-times slower than the bound (4.4). To see this, assume $\sigma \asymp n^{-\alpha}$ with $\alpha \in [0, \frac{1}{2}]$. If $\alpha \in [0, \frac{1}{2})$, then (4.4) implies the bound $O(n^{-\frac{1}{2}})$, while the bounds in [1, 28, 55] become $|R_S(A(S)) - R(A(S))| = O(n^{\alpha - \frac{1}{2}})$. For the special case $\alpha = 1/2$ suggested in the existing analysis of pointwise learning [51], Eq. (4.4) implies the bound $O(n^{-\frac{1}{2}} \log n)$, while the existing bound becomes $O(1)$ [1, 28, 55]. As we will clarify in Remark B.1, the existing stability analysis yields at best the excess risk bound $R(A(S)) - R(\mathbf{w}_R^*) = O(n^{-\frac{1}{4}})$ no matter how $\sigma$ changes. As a comparison, our stability analysis yields the bound $R(A(S)) - R(\mathbf{w}_R^*) = O(n^{-\frac{1}{2}} \log n)$.

**Remark 3** (Boundedness assumption). To get the bound $O(n^{-\frac{1}{2}} \log n)$, the existing stability analysis requires a boundedness assumption on loss functions as $0 \leq \ell(A(S); z, \tilde{z}) \leq B$ for a constant $B > 0$ and all $S \in \mathcal{Z}^n, z, z' \in \mathcal{Z}$ ($B$ is treated as a constant absorbed in a big O notation) [1, 8, 19, 20, 55] or a boundness assumption on $\mathcal{W}$ [28]. However, one can only show $\|A(S)\| = O(1/\sqrt{\sigma})$ [1] and therefore the constant $B$ needs to grow as $O(1/\sqrt{\sigma})$ for popular loss functions (e.g., hinge loss and logistic loss), from which the stability analysis in [8] can only imply suboptimal bounds $O((n\sigma)^{-1/2})$ even in the case of *pointwise learning* if one does not impose a boundedness assumption (note $\sigma$ is often very small). We develop the generalization bound $O(n^{-\frac{1}{2}} \log n)$ by relaxing the boundedness assumption to a variance assumption on $\ell(\mathbf{w}^*; Z, \tilde{Z})$. Note that the expectation of $\ell(\mathbf{w}^*; Z, \tilde{Z})$ is $R(\mathbf{w}^*)$, which is small according to the definition of $\mathbf{w}^*$. Therefore, it is reasonable to assume that the variance of $\ell(\mathbf{w}^*; Z, \tilde{Z})$ is bounded. To achieve this relaxation, we use a novel application of Theorem 1 to $\tilde{\ell}(\mathbf{w}; z, \tilde{z}) = \ell(\mathbf{w}; z, \tilde{z}) - \ell(\mathbf{w}^*; z, \tilde{z})$ instead of $\ell(\mathbf{w}; z, \tilde{z})$ (cf. Line 107 in the Appendix). Moreover, we introduce a novel lemma (Lemma 2) to show $\left|\mathbb{E}_S\left[\tilde{\ell}(A(S); z, \tilde{z})\right]\right| = O(1/(\sqrt{n}\sigma))$ (cf. Line 105 in the Appendix).

## 4.3 Generalization bounds for stochastic gradient descent

As a further application of Theorem 1, we establish generalization bounds for SGD (3.3) in pairwise learning, which can be considered as a deterministic algorithm if we fix $\{(i_t, j_t)_t\}$ in (3.3). SGD is a highly popular algorithm with wide applications in the big-data era. Note we do not require a strong convexity. We say $g : \mathcal{W} \mapsto \mathbb{R}$ is $\alpha$-smooth w.r.t. a norm $\|\cdot\|$ if $g$ is differentiable and

$$\|g'(\mathbf{w}) - g'(\mathbf{w}')\| \leq \alpha \|\mathbf{w} - \mathbf{w}'\|, \quad \forall \mathbf{w}, \mathbf{w}' \in \mathcal{W}.$$

Popular smooth loss functions include the logistic loss, the Huber loss, the squared hinge loss and the least squares loss [52]. Note that the logistic loss and the Huber loss are also Lipschitz continuous. The proof is given in Appendix C.

**Theorem 4.** *Let* (4.3) *hold. Assume for all* $z, z'$, $\mathbf{w} \mapsto \ell(\mathbf{w}; z, z')$ *is convex and* $\alpha$-*smooth w.r.t. the Euclidean norm, and for any* $\{(i_t, j_t)_t\}$ *we have* $\left|\mathbb{E}_S[\ell(\mathbf{w}_T; z, z')]\right| \leq M$, *where* $\mathbf{w}_T$ *is produced by SGD with* $\eta_t = c/\sqrt{T}, c \leq 2/\alpha$ *and* $r(\mathbf{w}) = 0$. *For any* $\delta \in (0, 1)$ *with probability* $1 - \delta$ *we have*

$$\left|R_S(\mathbf{w}_T) - R(\mathbf{w}_T)\right| = O\left(\log n \log(1/\delta)\sqrt{T}/n + n^{-\frac{1}{2}} \log n \log^{\frac{3}{2}}(1/\delta)\right). \tag{4.6}$$

**Remark 4.** We now show the implication of Theorem 4 on understanding the generalization behavior and implicit regularization of SGD. We can show (the details are given in Remark C.1)

$$R(\mathbf{w}_T) - R(\mathbf{w}_R^*) = \left(R(\mathbf{w}_T) - R_S(\mathbf{w}_T)\right) + \left(R_S(\mathbf{w}_T) - R_S(\mathbf{w}_R^*)\right) + O(n^{-\frac{1}{2}}). \tag{4.7}$$

The first term is the estimation error and the second term is the optimization error. Therefore, Theorem 4 actually gives an estimation error bound. If we further assume $\|\mathbf{w}_t\| \leq B$ for some $B > 0$ and all $t$, the optimization error was shown to satisfy[2] $R_S(\mathbf{w}_T) - R_S(\mathbf{w}_R^*) = O(T^{-\frac{1}{2}} \log T)$ [26] with high probability. Plugging these estimation and optimization error bounds back into (4.7), we derive with high probability $R(\mathbf{w}_T) - R(\mathbf{w}_R^*) = O\left(\log n \sqrt{T}/n + n^{-\frac{1}{2}} \log n\right) + O(T^{-\frac{1}{2}} \log T)$. It is clear that estimation errors increase as we run more iterations, while optimization errors decrease. One can take an optimal $T \asymp n$ to trade-off the optimization and estimation errors, and get

$R(\mathbf{w}_T) - R(\mathbf{w}_R^*) = O(n^{-\frac{1}{2}} \log n)$. To the best of our knowledge, this gives the first high-probability generalization bound for SGD in pairwise learning. Although we do not use an explicit regularizer in Theorem 4, our analysis shows that an implicit regularization can be achieved by tuning the number of iterations [25, 57]. We can compare our results with those based on the existing connection (4.2) between generalization and stability. Indeed, if we combine the best known optimization error bounds [26], the uniform stability of SGD established in Lemma C.3 and (4.2) together, we can only derive vacuous excess risk bound $R(\mathbf{w}_T) - R(\mathbf{w}_R^*) = O(1)$ (details are given in Remark C.2), which are significantly improved to $O(n^{-\frac{1}{2}} \log n)$ based on Theorem 1. In Appendix F we show $O(n^{-\frac{1}{2}})$ is minimax optimal for pairwise learning in a general convex setting.

The authors of [48] studied a variant of SGD where the models are updated by

$$\mathbf{w}_{t+1} = \mathbf{w}_t - \frac{\eta_t}{t-1} \sum_{k=1}^{t-1} \ell'(\mathbf{w}_t; z_{i_t}, z_{i_k}), \quad \forall t > 1.$$

Their stability bounds were stated in expectation [48] while we give high-probability analysis.

It should be mentioned that our generalization analysis can be applied to other iterative algorithms for pairwise learning, including gradient descent, Nesterov's accelerated gradient descent, the heavy ball method and stochastic gradient Langevin dynamics [11]. To this aim, it suffices to estimate the uniform stability of these algorithms in pairwise learning.

### 4.4 Optimistic generalization bounds

Our key idea to derive optimistic bounds is to use the on-average stability in Definition 2, whose connection to generalization is established in the following theorem to be proved in Section D.

**Theorem 5.** *If $A$ is $\gamma$-on-average stable, then $\mathbb{E}\big[R(A(S)) - R_S(A(S))\big] \leq \gamma$.*

We now present an optimistic generalization bound for pairwise learning by exploiting the smoothness of loss functions. By optimistic we mean that the decay rate of generalization bounds depends on the behavior of the best model. That is, we can get faster bounds if $R(\mathbf{w}_R^*) = o(1)$. Optimistic bounds were studied for pointwise learning in the literature [43, 50, 60], which are becoming interesting in the big-data era where models are often powerful enough to achieve a very small training error. For any $\mathbf{w} \in \mathcal{W}$, let $F(\mathbf{w}) = R(\mathbf{w}) + r(\mathbf{w})$. Theorem 6 is proved in Appendix D.

**Theorem 6.** *Assume for all $z, z'$, the map $\mathbf{w} \mapsto \ell(\mathbf{w}; z, z')$ is $\alpha$-smooth w.r.t. $\|\cdot\|$. Let $A(S) = \arg\min_{\mathbf{w} \in \mathcal{W}} F_S(\mathbf{w})$ and $\sigma n \geq 8\alpha$. If for all $S \in \mathcal{Z}^n$, $F_S$ is $\sigma$-strongly convex w.r.t. $\|\cdot\|$, then*

$$\mathbb{E}\big[F(A(S))\big] - F(\mathbf{w}^*) \leq \mathbb{E}\big[R(A(S)) - R_S(A(S))\big] \leq \Big(\frac{1024\alpha^2}{n^2\sigma^2} + \frac{64\alpha}{n\sigma}\Big)\mathbb{E}\big[R_S(A(S))\big]. \quad (4.8)$$

*Furthermore, if $r(\mathbf{w}) = O(\sigma\|\mathbf{w}\|^2)$ we can take some appropriate $\sigma$ to get*

$$\mathbb{E}[R(A(S))] - R(\mathbf{w}_R^*) = O\big(\sqrt{R(\mathbf{w}_R^*)}\|\mathbf{w}_R^*\|n^{-\frac{1}{2}} + \|\mathbf{w}_R^*\|^2 n^{-1}\big). \quad (4.9)$$

Note if $R(\mathbf{w}_R^*) = O(\|\mathbf{w}_R^*\|^2/n)$, the above excess risk bound becomes $\mathbb{E}[R(A(S))] - R(\mathbf{w}_R^*) = O(\|\mathbf{w}_R^*\|^2/n)$. That is, we get a fast excess risk bound if there exists a model with a small population risk.

## 5 Applications

### 5.1 Ranking

For ranking we assume real-valued labels indicating a ranking preference on instances, i.e., $y_i < y_j$ means $x_i$ has a lower rank than $x_j$. We aim to build a function $h_{\mathbf{w}} : \mathcal{X} \mapsto \mathbb{R}$ that ranks instances with larger labels higher than those with smaller labels [1, 13, 44]. The performance of a model $h_{\mathbf{w}}$ at a pair $z, z'$ can be measured by the 0-1 loss $\ell_{0\text{-}1}(\mathbf{w}; z, z') = \mathbb{I}[\text{sgn}(y - y')(h_{\mathbf{w}}(x) - h_{\mathbf{w}}(x')) < 0]$, where $\mathbb{I}[\cdot]$ is the indicator function taking the value 1 if the argument holds and 0 otherwise, and $\text{sgn}(a)$ denotes the sign of the number $a$. Since the 0-1 loss leads to an NP-hard problem, we consider loss functions of the form $\ell^\psi(\mathbf{w}; z, z') = \psi(\text{sgn}(y - y')(h_{\mathbf{w}}(x) - h_{\mathbf{w}}(x')))$. Here $\psi : \mathbb{R} \mapsto \mathbb{R}_+$ is

convex and decreasing, for which popular choices include the hinge loss $\psi(t) = \max\{1 - t, 0\}$ and the logistic loss $\psi(t) = \log(1 + \exp(-t))$. Below we provide bounds for RRM, SGD and optimistic bounds.

**Regularized risk minimization**. The following proposition follows directly from Theorem 3 by noticing that $r(\mathbf{w}) = \lambda\|\cdot\|^2$ is $2\lambda$-strongly convex w.r.t. $\|\cdot\|$. We omit the proof for simplicity.

**Proposition 7.** *Let Assumption 1 hold for both* $\mathbf{w}^*$ *and* $\mathbf{w}^*$ *replaced by* $\mathbf{w}_R^*$. *Consider ranking problems with* $f(\mathbf{w}; z, z') = \ell^\psi(\mathbf{w}; z, z') + \lambda\|\mathbf{w}\|^2$. *Assume* $\ell^\psi$ *is convex w.r.t.* $\mathbf{w}$ *and satisfy* (4.3). *Then for* $A(S) = \arg\min_{\mathbf{w}\in\mathcal{W}} F_S(\mathbf{w})$, $\lambda \asymp n^{-\frac{1}{2}}$ *and any* $\delta \in (0, 1/e)$, *with probability at least* $1 - \delta$ *there holds* $R(A(S)) - R(\mathbf{w}_R^*) = O\left(n^{-\frac{1}{2}}\log n\log(1/\delta)\right)$.

**Remark 5.** High-probability bounds for ranking were developed in the literature under some capacity assumptions on the hypothesis space $\{h_\mathbf{w} : \mathbf{w} \in \mathcal{W}\}$ measured by either covering numbers [44, 58] or VC dimension [13]. The arguments there are based on the uniform convergence of empirical risks to population risk and ignore the specific property of the learning algorithm, which inevitably depends on the complexity of the hypothesis space. Furthermore, a dependency on the dimension is necessary if no structural assumptions are imposed [18]. For example, generalization bounds $O(\sqrt{d/n})$ were derived for bipartite ranking (AUC maximization) via the uniform convergence approach [2]. As a comparison, we derive dimension-independent bounds of order $O(n^{-\frac{1}{2}}\log n)$. Furthermore, the existing stability analysis implies $|R_S(A(S)) - R(A(S))| = O\left(\lambda^{-1}\sqrt{1/n}\right)$ [1, 15] and the excess risk bounds $R(A(S)) - R(\mathbf{w}_R^*) = O(n^{-\frac{1}{4}})$, which are worse than the results in Proposition 7.

**Stochastic gradient descent**. The following proposition is a direct application of Theorem 4.

**Proposition 8.** *Consider ranking problems with* $f(\mathbf{w}; z, z') = \ell^\psi(\mathbf{w}; z, z')$, *i.e.* $r(\mathbf{w}) = 0$. *Let* (4.3) *hold and assume for all* $z, z'$, *the map* $\mathbf{w} \mapsto \ell^\psi(\mathbf{w}; z, z')$ *is convex and* $\alpha$-*smooth. Let* $\mathbf{w}_T$ *be produced by SGD with* $\eta_t = c/\sqrt{T}, c \le 2/\alpha$ *and assume* $\left|\mathbb{E}_S[\ell(\mathbf{w}_T; z, z')]\right| \le M$. *Then for any* $\delta \in (0, 1)$ *and* $T \asymp n$, *with probability* $1 - \delta$ *we have* $\left|R_S(\mathbf{w}_T) - R(\mathbf{w}_T)\right| = O\left(n^{-\frac{1}{2}}\log n\log^{\frac{3}{2}}(1/\delta)\right)$.

**Optimistic bounds**. Proposition 9 on optimistic bounds is a direct application of Theorem 6.

**Proposition 9.** *Consider ranking problems with* $f(\mathbf{w}; z, z') = \ell^\psi(\mathbf{w}; z, z') + \lambda\|\mathbf{w}\|^2$. *If* $\ell^\psi$ *is convex,* $\alpha$-*smooth and* $A(S) = \arg\min_{\mathbf{w}\in\mathcal{W}} F_S(\mathbf{w})$, *we can choose some* $\lambda \ge 8\alpha/n$ *such that* (4.9) *holds.*

Our results directly apply to bipartite ranking (AUC maximization) [14] with $\mathcal{Y} = \{+1, -1\}$. To see this, bipartite ranking is a specific instance of (3.1) with loss functions of the form $\ell^\psi(\mathbf{w}; z, z') = \psi((h_\mathbf{w}(x) - h_\mathbf{w}(x')))\mathbb{I}[y = 1, y' = -1]$. We omit this discussion for brevity.

## 5.2 Metric learning

We consider metric learning for learning a metric to measure the distance between instance pairs. We consider supervised metric learning with $\mathcal{Y} = \{-1, +1\}$, where we want an instance pair to be similar if they have the same class label, and apart from each other if they have different class labels [9, 28, 58]. We consider the Mahalanobis metric $h_\mathbf{w}(x, x') = \langle \mathbf{w}, (x - x')(x - x')^\top \rangle$, where $x^\top$ denotes the transpose of $x$ and $\mathbf{w} \in \mathbb{R}^{d\times d}$. The performance of $h_\mathbf{w}$ on $z, z'$ can be measured by the 0-1 loss $\ell_{0\text{-}1}(\mathbf{w}; z, z') = \mathbb{I}[\tau(y, y')(1 - h_\mathbf{w}(x, x')) \le 0]$, where $\tau(y, y') = 1$ if $y = y'$ and $-1$ otherwise. We often use a convex surrogate $\psi : \mathbb{R} \mapsto \mathbb{R}_+$, which leads to $\ell^\psi(\mathbf{w}; z, z') = \psi(\tau(y, y')(1 - h_\mathbf{w}(x, x')))$. We assume $\sup_{x\in\mathcal{X}}\|x\|_2 \le B$ for some $B > 0$, where $\|\cdot\|_2$ is the Euclidean norm.

**Regularized risk minimization**. Corollary 10 on RRM is proved in Appendix E.

**Corollary 10.** *Let Assumption 1 hold for both* $\mathbf{w}^*$ *and* $\mathbf{w}^*$ *replaced by* $\mathbf{w}_R^*$. *Consider metric learning with* $\mathcal{Y} = \{-1, +1\}$. *Consider* $f(\mathbf{w}; z, z') = \psi(\tau(y, y')(1 - h_\mathbf{w}(x, x'))) + \lambda\|\mathbf{w}\|^2$, *where* $\|\cdot\|$ *is the Frobenius norm and* $\psi(t) = \max\{0, 1 - t\}$. *Then for* $A(S) = \arg\min_{\mathbf{w}\in\mathcal{W}} F_S(\mathbf{w})$, $\lambda \asymp n^{-\frac{1}{2}}$ *and any* $\delta \in (0, 1/e)$, *with probability* $1 - \delta$ *it holds* $R(A(S)) - R(\mathbf{w}_R^*) = O\left(n^{-\frac{1}{2}}\log n\log(1/\delta)\right)$.

**Remark 6.** We make some comparisons. It was previously shown that $R(A(S)) - R_S(A(S)) = O\left(n^{-\frac{1}{2}}\lambda^{-1}\right)$ [9, 28, 55], which leads to the excess risk bound $O(n^{-\frac{1}{4}})$. This is significantly improved to $O(n^{-\frac{1}{2}}\log n)$ in Corollary 10. A uniform convergence rate $O(\sqrt{d n^{-1}})$ was shown for metric learning [54], which is not appealing for high-dimensional problems. It was further indicated that a

strong dependence on $d$ is generally necessary for the uniform convergence if one does not impose a structural assumption [54]. As a comparison, our bound in Corollary 10 is dimension-independent.

**Stochastic gradient descent**. Corollary 11 on bounds for SGD is proved in Appendix E.

**Corollary 11.** *Consider metric learning with $\mathcal{Y} = \{-1, +1\}$ and $f(\mathbf{w}; z, z') = \psi(\tau(y, y')(1 - h_{\mathbf{w}}(x, x')))$, where $\psi(t) = \log(1 + \exp(-t))$. Let $\mathbf{w}_T$ be produced by SGD with $\eta_t = c/\sqrt{T}, c \leq 1/8B^4$ and assume $\left|\mathbb{E}_S[\psi(\tau(y, y')(1 - h_{\mathbf{w}_T}(x, x')))]\right| \leq M$. Then for any $\delta \in (0, 1)$ and $T \asymp n$, with probability $1 - \delta$ we have $\left|R_S(\mathbf{w}_T) - R(\mathbf{w}_T)\right| = O\left(n^{-\frac{1}{2}} \log n \log^{\frac{3}{2}}(1/\delta)\right)$.*

**Optimistic bounds**. We also get optimistic bounds for metric learning. We omit the proof for brevity.

**Corollary 12.** *Let Assumptions of Corollary 10 hold except that we consider the logistic loss $\psi(t) = \log(1 + \exp(-t))$. If $A(S) = \arg\min_{\mathbf{w} \in \mathcal{W}} F_S(\mathbf{w})$ and $\|\mathbf{w}_R^*\| = O(1)$, then we can choose some appropriate $\lambda$ such that Eq. (4.9) holds.*

## 6   Conclusion

We analyze the generalization ability of pairwise learning using the methodology of algorithmic stability. We significantly improve the existing high-probability bounds $O(\sqrt{n}\gamma)$ to $O(\gamma \log n)$ for $\gamma$-uniformly stable algorithms. This allows us to improve the previously best excess risk bounds $O(n^{-1/4})$ for RRM and $O(1)$ for SGD to $O(n^{-1/2} \log n)$. As compared to the uniform convergence analysis, our stability analysis implies the first high-probability risk bound for SGD in pairwise learning, and yields bounds independent of the complexity of models and the input dimension. Furthermore, we introduce an on-average stability to develop optimistic bounds as fast as $O(1/n)$ for learning in a low noise setting. Specific applications are further given to show the advantage of our generalization bounds over the existing analysis.

Below we mention some interesting directions for future research. First, it would be interesting to extend the analysis here to other learning settings, such as distributed learning and online learning for pairwise learning. Second, one could tackle the challenging problem of stability and generalization bounds for non-convex pairwise learning problems, which are popular in modern machine learning.

## Broader Impact

This work does not present any foreseeable societal consequence.

## Acknowledgments and Disclosure of Funding

YL acknowledges support by the National Natural Science Foundation of China (Grant Nos. 61806091, 11771012), and by the Alexander von Humboldt Foundation for a Humboldt Research Fellowship. AL and MK acknowledge support by the German Research Foundation (DFG) award KL 2698/2-1 and by the Federal Ministry of Science and Education (BMBF) awards 01IS18051A and 031B0770E.

## Footnotes

*The first two authors contributed equally

[2]Although they considered step size $\eta_t = c/\sqrt{t}$ [26], their result also holds for $\eta_t = c/\sqrt{T}$.

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
