[Supplementary Material]

# Sharper Generalization Bounds for Pairwise Learning: Supplementary Material

## A Proof of Theorem 1

To prove Theorem 1, we need to introduce some lemmas. The following lemma is attributed to [7], which provides far-reaching moment bounds for a summation of weakly dependent and mean-zero random functions with bounded increments under a change of any single coordinate. We denote $S\backslash\{z_i\}$ the set $\{z_1, \ldots, z_{i-1}, z_{i+1}, \ldots, z_n\}$. The $L_p$-norm of a random variable $Z$ is denoted by $\|Z\|_p := \left(\mathbb{E}[|Z|^p]\right)^{\frac{1}{p}}, p \geq 1$.

**Lemma A.1** ([4]). *Let $S = \{z_1, \ldots, z_n\}$ be a set of independent random variables each taking values in $\mathcal{Z}$ and $M > 0$. Let $g_1, \ldots, g_n$ be some functions $g_i : \mathcal{Z}^n \mapsto \mathbb{R}$ such that the following holds for any $i \in [n]$*

- *$\left|\mathbb{E}_{S\backslash\{z_i\}}[g_i(S)]\right| \leq M$ almost surely (a.s.),*

- *$\mathbb{E}_{z_i}\left[g_i(S)\right] = 0$ a.s.,*

- *for any $j \in [n]$ with $j \neq i$, and $z_j'' \in \mathcal{Z}$*

$$\left|g_i(S) - g_i(z_1, \ldots, z_{j-1}, z_j'', z_{j+1}, \ldots, z_n)\right| \leq \beta. \tag{A.1}$$

*Then, for any $p \geq 2$*

$$\left\|\sum_{i=1}^{n} g_i(S)\right\|_p \leq 12\sqrt{6}pn\beta\lceil\log_2 n\rceil + 3\sqrt{2}M\sqrt{pn}.$$

The bounds on moments of random variables can be used to establish concentration inequalities, as shown in the following lemma [4, 16].

**Lemma A.2.** *Let $a, b \in \mathbb{R}_+$ and $\delta \in (0, 1/e)$. Let $Z$ be a random variable with $\|Z\|_p \leq \sqrt{p}a + pb$ for any $p \geq 2$. Then with probability at least $1 - \delta$*

$$|Z| \leq e\left(a\sqrt{\log(e/\delta)} + b\log(e/\delta)\right).$$

The following lemma controls the change on the output of stable algorithms if we perturb a training dataset by two examples.

**Lemma A.3.** *Let $A : \mathcal{Z}^n \mapsto \mathcal{W}$ be $\gamma$-uniformly stable. Then for any $S' = \{z_1', \ldots, z_n'\}$ and $i \neq j$, we have*

$$\sup_{z, \tilde{z} \in \mathcal{Z}} \left|\ell(A(S); z, \tilde{z}) - \ell(A(S_{i,j}); z, \tilde{z})\right| \leq 2\gamma,$$

*where $S_{i,j}$ is defined in* (3.4).

*Proof.* For any $i \in [n]$, introduce

$$S_i = \{z_1, \ldots, z_{i-1}, z_i', z_{i+1}, \ldots, z_n\}. \tag{A.2}$$

Note that $S, S_i$ differ only by a single example, and $S_i, S_{i,j}$ differ only by a single example. It then follows from the definition of uniform stability that

$$\sup_{z,\tilde{z}\in\mathcal{Z}} \left|\ell(A(S);z,\tilde{z}) - \ell(A(S_{i,j});z,\tilde{z})\right|$$

$$\leq \sup_{z,\tilde{z}\in\mathcal{Z}} \left|\ell(A(S);z,\tilde{z}) - \ell(A(S_i);z,\tilde{z})\right| + \sup_{z,\tilde{z}\in\mathcal{Z}} \left|\ell(A(S_i);z,\tilde{z}) - \ell(A(S_{i,j});z,\tilde{z})\right|$$

$$\leq 2\gamma.$$

The proof is complete. $\qquad\qquad\qquad\qquad\qquad\qquad\qquad\qquad\qquad\qquad\qquad\qquad\quad\square$

With these lemmas, we can give the proof of Theorem 1 on high-probability bounds of the generalization gap. The concentration inequality established in Lemma A.1 applies to a summation of $n$ random functions involving $n$ independent random variables, which does not apply to the objective function in pairwise learning since it is a $U$-statistic. We introduce a novel decomposition to exploit the structure of pairwise learning problems. We abbreviate $\sum_{i,j\in[n]:i\neq j}$ as $\sum_{i\neq j}$.

*Proof of Theorem 1.* Let $p \geq 2$ be any number. We can decompose the generalization gap associated to $A(S)$ as follows

$$n(n-1)\mathbb{E}_{Z,\tilde{Z}}\left[\ell(A(S);Z,\tilde{Z})\right] - \sum_{i\neq j}\ell(A(S);z_i,z_j) = \sum_{i\neq j}\mathbb{E}_{Z,\tilde{Z}}\left[\ell(A(S);Z,\tilde{Z}) - \mathbb{E}_{z_i',z_j'}[\ell(A(S_{i,j});Z,\tilde{Z})]\right]$$

$$+ \sum_{i\neq j}\mathbb{E}_{z_i',z_j'}\left[\mathbb{E}_{Z,\tilde{Z}}\left[\ell(A(S_{i,j});Z,\tilde{Z})\right] - \ell(A(S_{i,j});z_i,z_j)\right] + \sum_{i\neq j}\mathbb{E}_{z_i',z_j'}\left[\ell(A(S_{i,j});z_i,z_j) - \ell(A(S);z_i,z_j)\right],$$

where $S_{i,j}$ is defined in (3.4). According to Lemma A.3, we know

$$\left|\ell(A(S);Z,\tilde{Z}) - \mathbb{E}_{z_i',z_j'}[\ell(A(S_{i,j});Z,\tilde{Z})]\right| \leq 2\gamma$$

and

$$\left|\ell(A(S_{i,j});z_i,z_j) - \ell(A(S);z_i,z_j)\right| \leq 2\gamma.$$

Therefore,

$$\left|n(n-1)\mathbb{E}_{Z,\tilde{Z}}\left[\ell(A(S);Z,\tilde{Z})\right] - \sum_{i\neq j}\ell(A(S);z_i,z_j)\right| \leq 4n(n-1)\gamma + \left|\sum_{i\neq j}g_{i,j}(S)\right|, \quad \text{(A.3)}$$

where we introduce

$$g_{i,j}(S) = \mathbb{E}_{z_i',z_j'}\left[\mathbb{E}_{Z,\tilde{Z}}\left[\ell(A(S_{i,j});Z,\tilde{Z})\right] - \ell(A(S_{i,j});z_i,z_j)\right], \quad \forall i,j\in[n].$$

For any $i,j\in[n]$, we can further decompose $g_{i,j}$ as $g_{i,j} = g_j^{(i)} + \tilde{g}_i^{(j)}$, where (we omit the argument $S$ for brevity)

$$g_j^{(i)} = \mathbb{E}_{z_i',z_j'}\left[\mathbb{E}_{Z,\tilde{Z}}\left[\ell(A(S_{i,j});Z,\tilde{Z})\right] - \mathbb{E}_Z[\ell(A(S_{i,j});Z,z_j)]\right]$$

$$\tilde{g}_i^{(j)} = \mathbb{E}_{z_i',z_j'}\left[\mathbb{E}_Z\left[\ell(A(S_{i,j});Z,z_j] - \ell(A(S_{i,j});z_i,z_j)\right].$$

Let us temporarily fix $i$, and consider $n-1$ random functions $g_1^{(i)}, \ldots, g_{i-1}^{(i)}, g_{i+1}^{(i)}, \ldots, g_n^{(i)}$. According to the assumption $\left|\mathbb{E}_S[\ell(A(S);z,\tilde{z})]\right| \leq M$ for all $z, \tilde{z}$, we know

$$\left|\mathbb{E}_{S\setminus\{z_j\}}[g_j^{(i)}(S)]\right| \leq 2M, \quad \forall j\in[n].$$

For any $j \neq i$, since $z_j$ is independent of $S_{i,j}$ we know

$$\mathbb{E}_{z_j}\left[\mathbb{E}_{Z,\tilde{Z}}\left[\ell(A(S_{i,j});Z,\tilde{Z})\right] - \mathbb{E}_Z[\ell(A(S_{i,j});Z,z_j)]\right] = 0.$$

Therefore, $\mathbb{E}_{z_j}[g_j^{(i)}] = 0$. For any $k \neq j$ and any $z_k'' \in \mathcal{Z}$, it is clear from the uniform stability of $A$ that

$$\left|\mathbb{E}_{z_i',z_j'}\mathbb{E}_{Z,\tilde{Z}}\left[\ell(A(S_{i,j});Z,\tilde{Z})\right] - \mathbb{E}_{z_i',z_j'}\mathbb{E}_{Z,\tilde{Z}}\left[\ell(A(S_{i,j}^{(k)});Z,\tilde{Z})\right]\right| \leq \gamma,$$

where $S_{i,j}^{(k)}$ is the set derived by replacing the $k$-th element of $S_{i,j}$ with $z_k''$. Similarly, one have

$$\left| \mathbb{E}_{z_i',z_j'} \mathbb{E}_Z[\ell(A(S_{i,j}); Z, z_j)] - \mathbb{E}_{z_i',z_j'} \mathbb{E}_Z[\ell(A(S_{i,j}^{(k)}); Z, z_j)] \right| \leq \gamma.$$

It then follows from the above two inequalities that $g_j^{(i)}$ satisfies the bounded increment condition (A.1) with $\beta = 2\gamma$ for all $k \neq j$, i.e.,

$$\left| \mathbb{E}_{z_i',z_j'} \Big[ \mathbb{E}_{Z,\tilde{Z}} \big[\ell(A(S_{i,j}); Z, \tilde{Z})\big] - \mathbb{E}_Z[\ell(A(S_{i,j}); Z, z_j)]\Big] \right.$$
$$\left. - \mathbb{E}_{z_i',z_j'} \Big[ \mathbb{E}_{Z,\tilde{Z}} \big[\ell(A(S_{i,j}^{(k)}); Z, \tilde{Z})\big] - \mathbb{E}_Z[\ell(A(S_{i,j}^{(k)}); Z, z_j)]\Big] \right| \leq 2\gamma.$$

Therefore, all the assumptions of Lemma A.1 hold for the random functions $g_1^{(i)}, \ldots, g_{i-1}^{(i)}, g_{i+1}^{(i)}, \ldots, g_n^{(i)}$ with $n$ there replaced by $n-1$ and $\beta = 2\gamma$. We can apply Lemma A.1 to derive

$$\left\| \sum_{j\in[n],j\neq i} g_j^{(i)} \right\|_p \leq 24\sqrt{6}p(n-1)\gamma\lceil \log_2(n-1) \rceil + 6\sqrt{2}M\sqrt{p(n-1)}, \quad \forall i \in [n].$$

Similarly, we can also show that

$$\left\| \sum_{i\in[n],i\neq j} \tilde{g}_i^{(j)} \right\|_p \leq 24\sqrt{6}p(n-1)\gamma\lceil \log_2(n-1) \rceil + 6\sqrt{2}M\sqrt{p(n-1)}, \quad \forall j \in [n].$$

It then follows from the subadditivity of $\|\cdot\|_p$ and the above two inequalities that

$$\left\| \sum_{i\neq j} g_{i,j} \right\|_p \leq \left\| \sum_{i\neq j} g_j^{(i)} \right\|_p + \left\| \sum_{i\neq j} \tilde{g}_i^{(j)} \right\|_p$$
$$\leq \sum_{i\in[n]} \left\| \sum_{j\in[n],j\neq i} g_j^{(i)} \right\|_p + \sum_{j\in[n]} \left\| \sum_{i\in[n],i\neq j} \tilde{g}_i^{(j)} \right\|_p$$
$$\leq 48\sqrt{6}p(n-1)n\gamma\lceil \log_2(n-1) \rceil + 12\sqrt{2}M\sqrt{p(n-1)}n.$$

We can combine the above $p$-norm and Lemma A.2 to derive the following inequality with probability at least $1-\delta$

$$\left| \sum_{i\neq j} g_{i,j} \right| \leq e\Big(12\sqrt{2}M\sqrt{(n-1)}n\sqrt{\log(e/\delta)} + 48\sqrt{6}(n-1)n\gamma\lceil \log_2(n-1) \rceil \log(e/\delta)\Big).$$

Plugging the above inequality back into (A.3) and using the definition of $R_S, R$, we derive the following inequality with probability at least $1-\delta$

$$|R_S(A(S)) - R(A(S))| \leq 4\gamma + \frac{1}{n(n-1)} \Big| \sum_{i\neq j} g_{i,j} \Big|$$
$$\leq 4\gamma + e\Big(12\sqrt{2}M(n-1)^{-\frac{1}{2}}\sqrt{\log(e/\delta)} + 48\sqrt{6}\gamma\lceil \log_2(n-1) \rceil \log(e/\delta)\Big).$$

The proof is complete. $\qquad\square$

## B  Proof of Theorem 3

In this section, we prove Theorem 3 on high-probability bounds for learning with strongly convex objective functions. We first prove Lemma 2 on the norm of output model.

*Proof of Lemma 2.* Since $A(S)$ is the minimizer of $F_S$, we know there is a $F_S'(A(S)) = 0$ ($F_S'$ is a subgradient of $F_S$ at $A(S)$). This together with the definition of strong convexity implies

$$R_S(\mathbf{w}^*) + r(\mathbf{w}^*) - R_S(A(S)) - r(A(S)) \geq \frac{\sigma}{2}\|A(S) - \mathbf{w}^*\|^2. \tag{B.1}$$

Analogous to (A.3), we know

$$n(n-1)\Big(R(A(S)) - R_S(A(S))\Big) \leq 4n(n-1)\gamma + \sum_{i,j\in[n]:i\neq j} g_{i,j},$$

where $g_{i,j}$ is defined in the proof of Theorem 1. In the proof of Theorem 1, we have shown $\mathbb{E}[g_{i,j}] = 0$. It then follows that

$$\mathbb{E}\big[R(A(S)) - R_S(A(S))\big] \leq 4\gamma.$$

We can plug the above inequality back into (B.1) to derive

$$\frac{\sigma}{2}\mathbb{E}\big[\|A(S) - \mathbf{w}^*\|^2\big] \leq \mathbb{E}\big[R_S(\mathbf{w}^*) + r(\mathbf{w}^*) - R_S(A(S)) - r(A(S))\big]$$

$$\leq \mathbb{E}\big[R_S(\mathbf{w}^*) + r(\mathbf{w}^*) - R(A(S)) - r(A(S))\big] + 4\gamma$$

$$= \mathbb{E}\big[R(\mathbf{w}^*) + r(\mathbf{w}^*) - R(A(S)) - r(A(S))\big] + 4\gamma \leq 4\gamma,$$

where the last inequality holds since $\mathbf{w}^*$ minimizes $F = R + r$. The stated inequality then follows and finishes the proof. $\square$

To prove Theorem 3, we introduce some lemmas.

**Lemma B.1.** *For any* $S \in \mathcal{Z}^n$, *define* $A$ *as* $A(S) = \arg\min_{\mathbf{w}\in\mathcal{W}} F_S(\mathbf{w})$. *For any* $k \in [n]$, *let* $S_k$ *be defined by* (A.2). *Then*

$$F_S(A(S_k)) - F_S(A(S)) \leq$$
$$\frac{1}{n(n-1)} \sum_{i\in[n]:i\neq k} \Big(\big(\ell(A(S_k); z_i, z_k) - \ell(A(S); z_i, z_k)\big) + \big(\ell(A(S_k); z_k, z_i) - \ell(A(S); z_k, z_i)\big)$$

$$+ \big(\ell(A(S); z_i, z_k') - \ell(A(S_k); z_i, z_k')\big) + \big(\ell(A(S); z_k', z_i) - \ell(A(S_k); z_k', z_i)\big)\Big).$$

*Proof.* Without loss of generality, we can assume $k = n$. Since $A(S_n)$ is a minimizer of $F_{S_n}$, we know

$$F_S(A(S_n)) - F_S(A(S))$$
$$= F_S(A(S_n)) - F_{S_n}(A(S_n)) + F_{S_n}(A(S_n)) - F_{S_n}(A(S)) + F_{S_n}(A(S)) - F_S(A(S))$$
$$\leq F_S(A(S_n)) - F_{S_n}(A(S_n)) + F_{S_n}(A(S)) - F_S(A(S)). \tag{B.2}$$

By the definition of $F_S$ and $F_{S_n}$, we know

$$n(n-1)\big(F_S(A(S_n)) - F_{S_n}(A(S_n))\big) = \sum_{i,j\in[n]:i\neq j} f(A(S_n); z_i, z_j)$$

$$- \Big(\sum_{i,j\in[n-1]:i\neq j} f(A(S_n); z_i, z_j) + \sum_{i\in[n-1]} f(A(S_n); z_i, z_n') + \sum_{i\in[n-1]} f(A(S_n); z_n', z_i)\Big)$$

$$= \sum_{i\in[n-1]} \Big(f(A(S_n); z_i, z_n) + f(A(S_n); z_n, z_i) - f(A(S_n); z_i, z_n') - f(A(S_n); z_n', z_i)\Big).$$

Similarly, we know

$$n(n-1)\big(F_{S_n}(A(S)) - F_S(A(S))\big) =$$
$$\sum_{i\in[n-1]} \Big(f(A(S); z_i, z_n') + f(A(S); z_n', z_i) - f(A(S); z_i, z_n) - f(A(S); z_n, z_i)\Big).$$

Therefore, we can combine the above two identities to derive

$$n(n-1)\big(F_S(A(S_n)) - F_{S_n}(A(S_n)) + F_{S_n}(A(S)) - F_S(A(S))\big) =$$
$$\sum_{i\in[n-1]} \Big(\big(f(A(S_n); z_i, z_n) - f(A(S); z_i, z_n)\big) + \big(f(A(S_n); z_n, z_i) - f(A(S); z_n, z_i)\big) +$$

$$\big(f(A(S); z_i, z_n') - f(A(S_n); z_i, z_n')\big) + \big(f(A(S); z_n', z_i) - f(A(S_n); z_n', z_i)\big)\Big).$$

This together with the structure of $f$ ($f = \ell + r$ with $r$ depending only on $\mathbf{w}$) implies

$$n(n-1)\big(F_S(A(S_n)) - F_{S_n}(A(S_n)) + F_{S_n}(A(S)) - F_S(A(S))\big) =$$
$$\sum_{i \in [n-1]} \Big( \big(\ell(A(S_n); z_i, z_n) - \ell(A(S); z_i, z_n)\big) + \big(\ell(A(S_n); z_n, z_i) - \ell(A(S); z_n, z_i)\big) +$$
$$\big(\ell(A(S); z_i, z_n') - \ell(A(S_n); z_i, z_n')\big) + \big(\ell(A(S); z_n', z_i) - \ell(A(S_n); z_n', z_i)\big)\Big).$$

Plugging the above identity back into (B.2), we derive

$$F_S(A(S_n)) - F_S(A(S))$$
$$\leq \frac{1}{n(n-1)} \sum_{i \in [n-1]} \Big( \big(\ell(A(S_n); z_i, z_n) - \ell(A(S); z_i, z_n)\big) + \big(\ell(A(S_n); z_n, z_i) - \ell(A(S); z_n, z_i)\big)$$
$$+ \big(\ell(A(S); z_i, z_n') - \ell(A(S_n); z_i, z_n')\big) + \big(\ell(A(S); z_n', z_i) - \ell(A(S_n); z_n', z_i)\big)\Big).$$

The proof is complete. $\qquad\square$

The following lemma establishes the uniform stability of pairwise learning with strongly convex objectives.

**Lemma B.2.** *Define $A$ as $A(S) = \arg\min_{\mathbf{w} \in \mathcal{W}} F_S(\mathbf{w})$. Suppose $F_S$ is $\sigma$-strongly convex w.r.t. $\|\cdot\|$. Assume for all $z, \tilde{z}$ we have (4.3). Then $A$ is $\frac{8L^2}{n\sigma}$-uniformly stable.*

*Proof.* Let $S, S'$ be two sets that differ by a single example and let $\mathbf{w}_S = A(S)$ and $\mathbf{w}_{S'} = A(S')$. Without loss of generality, we can assume $S' = \{z_1, \ldots, z_{n-1}, z_n'\}$, i.e., $S$ and $S'$ differ by the last example.

Since $\mathbf{w}_S$ is a minimizer of $F_S$ we know there is a subgradient $F_S'(\mathbf{w}_S) = 0$, which together with the $\sigma$-strong convexity of $F_S$, implies

$$F_S(\mathbf{w}_{S'}) - F_S(\mathbf{w}_S) \geq \frac{\sigma}{2}\|\mathbf{w}_{S'} - \mathbf{w}_S\|^2. \qquad (\text{B.3})$$

According to (4.3) and Lemma B.1, we know

$$F_S(\mathbf{w}_{S'}) - F_S(\mathbf{w}_S) \leq \frac{4(n-1)L\|\mathbf{w}_S - \mathbf{w}_{S'}\|}{n(n-1)}.$$

which, together with (B.3), implies

$$\|\mathbf{w}_S - \mathbf{w}_{S'}\| \leq \frac{8L}{n\sigma}.$$

This further together with (4.3) implies the $\frac{8L^2}{n\sigma}$-uniform stability of $A$. The proof is complete. $\qquad\square$

To obtain tight control on the term $R(\mathbf{w}^*) - R_S(\mathbf{w}^*)$, we will need a version of Bernstein's inequality for U-statistics. The following theorem is attributed to [10], and can be found in [5] (inequality A.1 on page 868), and in [12] (Theorem 2). A complete proof is provided in [13] (page 4).

**Lemma B.3** (Bernstein's inequality for U-Statistic [10, 13])**.** *Let $Z_1, \ldots, \ldots, Z_n$ be independent variables taking values in $\mathcal{Z}$ and $q : \mathcal{Z} \times \mathcal{Z} \mapsto \mathbb{R}$. Let $b = \sup_{z, \tilde{z}} |q(z, \tilde{z})|$ and $\sigma_0^2$ be the variance of $q(Z, \tilde{Z})$. Then for any $\delta \in (0, 1)$ with probability at least $1 - \delta$*

$$\left| \frac{1}{n(n-1)} \sum_{i,j \in [n]: i \neq j} q(Z_i, Z_j) - \mathbb{E}_{Z, \tilde{Z}}[q(Z, \tilde{Z})] \right| \leq \frac{2b \log(1/\delta)}{3\lfloor n/2 \rfloor} + \sqrt{\frac{2\sigma_0^2 \log(1/\delta)}{\lfloor n/2 \rfloor}}. \qquad (\text{B.4})$$

We now give the proof of Theorem 3.

*Proof of Theorem 3.* According to Lemma B.2, we know that $A$ is $\frac{8L^2}{n\sigma}$-uniformly stable. Using this together with Lemma 2 we derive $\mathbb{E}_S[\|\mathbf{w}^* - A(S)\|^2] \leq \frac{64L^2}{n\sigma^2}$ and therefore

$$\mathbb{E}_S[\|\mathbf{w}^* - A(S)\|] \leq \left(\mathbb{E}_S[\|\mathbf{w}^* - A(S)\|^2]\right)^{\frac{1}{2}} \leq \frac{8L}{\sqrt{n}\sigma}. \tag{B.5}$$

For any $\mathbf{w} \in \mathcal{W}$ and $z, \tilde{z}$, define

$$\tilde{\ell}(\mathbf{w}; z, \tilde{z}) = \ell(\mathbf{w}; z, \tilde{z}) - \ell(\mathbf{w}^*; z, \tilde{z}).$$

Then it is clear from Lemma B.2 that $A$ is also $\frac{8L^2}{n\sigma}$-uniformly stable when measured by the "loss" $\tilde{\ell}$, i.e., for any $S, S'$ differing by one example

$$\sup_{z,\tilde{z}} \left|\tilde{\ell}(A(S); z, \tilde{z}) - \tilde{\ell}(A(S'); z, \tilde{z})\right|$$

$$= \sup_{z,\tilde{z}} \left|\ell(A(S); z, \tilde{z}) - \ell(A(S'); z, \tilde{z}) - \ell(\mathbf{w}^*; z, \tilde{z}) + \ell(\mathbf{w}^*; z, \tilde{z})\right|$$

$$= \sup_{z,\tilde{z}} \left|\ell(A(S); z, \tilde{z}) - \ell(A(S'); z, \tilde{z})\right| \leq 8L^2/(n\sigma).$$

Furthermore, by the Lipschitz continuity (4.3) and (B.5), we know the following inequality for all $z, \tilde{z} \in \mathcal{Z}$

$$\left|\mathbb{E}_S\left[\tilde{\ell}(A(S); z, \tilde{z})\right]\right| = \left|\mathbb{E}_S\left[\ell(A(S); z, \tilde{z}) - \ell(\mathbf{w}^*; z, \tilde{z})\right]\right|$$

$$\leq L\mathbb{E}_S\left[\|\mathbf{w}^* - A(S)\|\right] \leq \frac{8L^2}{\sqrt{n}\sigma}.$$

We can now apply Theorem 1, with $\gamma = \frac{8L^2}{n\sigma}$, $M = 8L^2/(\sqrt{n}\sigma)$ and $\ell$ replaced by $\tilde{\ell}$, and show the following inequality with probability $1 - \delta/2$

$$\left|\frac{1}{n(n-1)} \sum_{i \neq j} \tilde{\ell}(A(S); z_i, z_j) - \mathbb{E}_{z,\tilde{z}}\left[\tilde{\ell}(A(S); z, \tilde{z})\right]\right| \leq \frac{32L^2}{n\sigma}$$

$$+ e\left(\frac{96\sqrt{2}L^2\sqrt{\log(2e/\delta)}}{\sqrt{n(n-1)}\sigma}\right) + \frac{384\sqrt{6}L^2\lceil\log_2 n\rceil\log(2e/\delta)}{n\sigma},$$

from which we derive the following inequality with probability $1 - \delta/2$

$$\left|R_S(A(S)) - R(A(S))\right| \leq \left|\frac{1}{n(n-1)} \sum_{i \neq j} \ell(\mathbf{w}^*; z_i, z_j) - \mathbb{E}_{z,\tilde{z}}\left[\ell(\mathbf{w}^*; z, \tilde{z})\right]\right|$$

$$+ \frac{32L^2}{n\sigma}\left(1 + 3\sqrt{\frac{2n\log(2e/\delta)}{n-1}} + 12\sqrt{6}\lceil\log_2 n\rceil\log(2e/\delta)\right). \tag{B.6}$$

By to the definition of $\mathbf{w}^*$ ($R'(\mathbf{w}^*) + r'(\mathbf{w}^*) = 0$), the $\sigma$-strong convexity and Assumption 1 ($0 \leq \ell(0; z, \tilde{z})$), we know

$$\frac{\sigma\|\mathbf{w}^*\|^2}{2} \leq R(0) + r(0) - R(\mathbf{w}^*) - r(\mathbf{w}^*) \Longrightarrow \|\mathbf{w}^*\| \leq \sqrt{\frac{2(R(0) + r(0))}{\sigma}}.$$

It then follows from the Lipschitz continuity (4.3) that

$$|\ell(\mathbf{w}^*; z, \tilde{z})| = \left|\ell(\mathbf{w}^*; z, \tilde{z}) - \ell(0; z, \tilde{z}) + \ell(0; z, \tilde{z})\right| \leq L\|\mathbf{w}^*\| + \sup_{z,\tilde{z}} \ell(0; z, \tilde{z})$$

$$\leq L\sqrt{\frac{2(R(0) + r(0))}{\sigma}} + \sup_{z,\tilde{z}} \ell(0; z, \tilde{z}).$$

According to Bernstein's inequality (B.4), we derive the following inequality with probability $1 - \delta/2$ that

$$\left|\frac{1}{n(n-1)} \sum_{i \neq j} \ell(\mathbf{w}^*; z_i, z_j) - \mathbb{E}_{z,\tilde{z}}[\ell(\mathbf{w}^*; z, \tilde{z})]\right| \leq$$

$$\frac{2\left(L\sqrt{2(R(0) + r(0))/\sigma} + b\right)\log(2/\delta)}{3\lfloor n/2\rfloor} + \sqrt{\frac{2\sigma_0^2\log(2/\delta)}{\lfloor n/2\rfloor}}. \tag{B.7}$$

Plugging the above inequality back into (B.6), we derive the following inequality with probability at least $1 - \delta$

$$\left|R_S(A(S)) - R(A(S))\right| \leq \frac{2\big(L\sqrt{2(R(0) + r(0))/\sigma} + b\big)\log(2/\delta)}{3\lfloor n/2 \rfloor} + \sqrt{\frac{2\sigma_0^2 \log(2/\delta)}{\lfloor n/2 \rfloor}}$$
$$+ \frac{32L^2}{n\sigma}\Big(1 + 3\sqrt{\frac{2n \log(2e/\delta)}{n-1}} + 12\sqrt{6}\lceil \log_2 n \rceil \log(2e/\delta)\Big).$$

The above inequality can be written as the stated bound (4.4).

We now turn to (4.5). According to the definition of $R$ and $F$, we can decompose the excess risk $R(A(S)) - R(\mathbf{w}_R^*)$ as follows

$$R(A(S)) - R(\mathbf{w}_R^*)$$
$$= R(A(S)) - R_S(A(S)) + R_S(\mathbf{w}_R^*) - R(\mathbf{w}_R^*) + R_S(A(S)) - R_S(\mathbf{w}_R^*)$$
$$= R(A(S)) - R_S(A(S)) + R_S(\mathbf{w}_R^*) - R(\mathbf{w}_R^*) + F_S(A(S)) - F_S(\mathbf{w}_R^*) + r(\mathbf{w}_R^*) - r(A(S))$$
$$\leq R(A(S)) - R_S(A(S)) + R_S(\mathbf{w}_R^*) - R(\mathbf{w}_R^*) + O(\sigma\|\mathbf{w}_R^*\|^2) - r(A(S)), \qquad (B.8)$$

where we have used the inequality $F_S(A(S)) \leq F_S(\mathbf{w}_R^*)$ due to the definition of $A(S)$ and the assumption $r(\mathbf{w}) = O(\sigma\|\mathbf{w}\|^2)$ in the last step. Analogous to (B.7), one can use Bernstein's inequality (Lemma B.3) to show with probability at least $1 - \delta/2$ that (under a very mild assumption $\sup_{z,z'} \ell(\mathbf{w}_R^*; z, z') = O(\sqrt{n})$)

$$R_S(\mathbf{w}_R^*) - R(\mathbf{w}_R^*) = O\Big(\frac{\log(1/\delta)}{\sqrt{n}} + \sqrt{\frac{\sigma_0^2 \log(1/\delta)}{n}}\Big). \qquad (B.9)$$

Plugging the above inequality and (4.4) back into (B.8) shows the following inequality with probability at least $1 - \delta$

$$R(A(S)) - R(\mathbf{w}_R^*) = O\Big((n\sigma)^{-1} \log n \log(1/\delta) + n^{-\frac{1}{2}} \log(1/\delta)\Big) + O(\sigma\|\mathbf{w}_R^*\|^2).$$

The stated bound (4.5) follows with $\sigma \asymp n^{-1/2}$. The proof is complete. $\qquad \square$

**Remark B.1.** We show here that the existing stability bound (eq. (4.2) with $\gamma = O(1/(n\sigma))$) [1, 6, 11, 17]

$$|R_S(A(S)) - R(A(S))| = O(\sigma^{-1} n^{-\frac{1}{2}}) \qquad (B.10)$$

yields at best the excess risk bound $R(A(S)) - R(\mathbf{w}_R^*) = O(n^{-\frac{1}{4}})$. Indeed, plugging (B.10) and (B.9) back into (B.8), we derive the following inequality with high probability

$$R(A(S)) - R(\mathbf{w}_R^*) = O(\sigma^{-1} n^{-\frac{1}{2}}) + O(\sigma).$$

We can balance the above two terms by taking $\sigma \asymp n^{-\frac{1}{4}}$ and get

$$R(A(S)) - R(\mathbf{w}_R^*) = O(n^{-\frac{1}{4}}).$$

## C  Proof of Theorem 4

To prove Theorem 4, we first introduce some lemmas. Lemma C.1 shows the non-expansiveness of the gradient-update operator, which plays a key role in establishing the stability of SGD. Lemma C.2 is a Chernoff's bound for a summation of independent Bernoulli random variables [2]. In this section, we let $\|\cdot\|_2$ be the Euclidean norm.

**Lemma C.1** ([8]). *Assume for all $z \in \mathcal{Z}$, the function $\mathbf{w} \mapsto \ell(\mathbf{w}; z, z')$ is convex and $\alpha$-smooth. Then for all $\eta \leq 2/\alpha$ and $z, z' \in \mathcal{Z}$ there holds*

$$\|\mathbf{w} - \eta\ell'(\mathbf{w}; z, z') - \mathbf{w}' + \eta\ell'(\mathbf{w}'; z, z')\|_2 \leq \|\mathbf{w} - \mathbf{w}'\|_2.$$

**Lemma C.2** (Chernoff's Bound). *Let $X_1, \ldots, X_T$ be independent random variables taking values in $\{0, 1\}$. Let $X = \sum_{t=1}^T X_t$ and $\mu = \mathbb{E}[X]$. Then for any $\epsilon \in (0, 1)$ with probability at least $1 - \exp\big(-\mu\epsilon^2/3\big)$ we have $X \leq (1 + \epsilon)\mu$.*

We now establish the uniform stability of SGD. The randomness of SGD can be characterized by $\{\{(i_t, j_t)_t\} : i_t, j_t \in [n], i_t \neq j_t\}$. Therefore, SGD can be considered as a deterministic algorithm if $\{\{(i_t, j_t)_t\} : i_t, j_t \in [n], i_t \neq j_t\}$ is fixed. For simplicity, we consider two datasets that differ by the last example. However, our discussion directly extends to the general case where two datasets differ by a single example. Notice that the Lipschitz continuity (4.3) implies $\|\ell'(\mathbf{w}; z, z')\|_2 \leq L$.

**Lemma C.3.** *Consider fixed $\{\{(i_t, j_t)_t\} : i_t, j_t \in [n], i_t \neq j_t\}$. Let $S = \{z_1, \ldots, z_n\}$ and $S' = \{z_1', \ldots, z_n'\}$ be two datasets that differ only by the last example, i.e., $z_i = z_i'$ if $i \in [n-1]$. Suppose for all $z, z' \in \mathcal{Z}$ the function $\mathbf{w} \mapsto \ell(\mathbf{w}; z, z')$ is convex, $\alpha$-smooth and $L$-Lipschitz w.r.t. $\|\cdot\|_2$. Let $\{\mathbf{w}_t\}, \{\mathbf{w}_t'\}$ be produced by SGD on $S$ and $S'$ respectively with $\eta_t \leq 2/\alpha$, i.e., (3.3) with $r(\mathbf{w}) = 0$. Then SGD with $t$ iterations is $\gamma$-uniformly stable with*

$$\gamma \leq 2L^2 \sum_{k=1}^{t} \eta_k \mathbb{I}[i_k = n \text{ or } j_k = n].$$

*Proof.* Let us consider two cases. We first consider the case $i_t \in [n-1]$ and $j_t \in [n-1]$. In this case, according to the SGD update (3.3) with $r(\mathbf{w}) = 0$ we know

$$\mathbf{w}_{t+1} - \mathbf{w}_{t+1}' = \mathbf{w}_t - \eta_t \ell'(\mathbf{w}_t; z_{i_t}, z_{j_t}) - \mathbf{w}_t' + \eta_t \ell'(\mathbf{w}_t'; z_{i_t}', z_{j_t}')$$
$$= \mathbf{w}_t - \eta_t \ell'(\mathbf{w}_t; z_{i_t}, z_{j_t}) - \mathbf{w}_t' + \eta_t \ell'(\mathbf{w}_t'; z_{i_t}, z_{j_t}).$$

It then follows from Lemma C.1 that

$$\|\mathbf{w}_{t+1} - \mathbf{w}_{t+1}'\|_2 \leq \|\mathbf{w}_t - \mathbf{w}_t'\|_2.$$

We now consider the case that either $i_t = n$ or $j_t = n$. In this case, we know

$$\|\mathbf{w}_{t+1} - \mathbf{w}_{t+1}'\|_2 = \left\|\mathbf{w}_t - \eta_t \ell'(\mathbf{w}_t; z_{i_t}; z_{j_t}) - \mathbf{w}_t' + \eta_t \ell'(\mathbf{w}_t'; z_{i_t}', z_{j_t}')\right\|_2$$
$$\leq \|\mathbf{w}_t - \mathbf{w}_t'\|_2 + \left\|\eta_t \ell'(\mathbf{w}_t'; z_{i_t}', z_{j_t}') - \eta_t \ell'(\mathbf{w}_t; z_{i_t}, z_{j_t})\right\|_2$$
$$\leq \|\mathbf{w}_t - \mathbf{w}_t'\|_2 + 2\eta_t L,$$

where we have used $\|\ell'(\mathbf{w}; z, z')\|_2 \leq L$ due to the $L$-Lipschitzness. As a combination of the above two cases, we derive

$$\|\mathbf{w}_{t+1} - \mathbf{w}_{t+1}'\|_2 \leq \|\mathbf{w}_t - \mathbf{w}_t'\|_2 + 2\eta_t L \mathbb{I}[i_t = n \text{ or } j_t = n],$$

where $\mathbb{I}[\cdot]$ is the indicator function taking $1$ if the argument holds and $0$ otherwise. Taking a summation of the above inequality gives ($\mathbf{w}_1 = \mathbf{w}_1'$)

$$\|\mathbf{w}_{t+1} - \mathbf{w}_{t+1}'\|_2 \leq 2L \sum_{k=1}^{t} \eta_k \mathbb{I}[i_k = n \text{ or } j_k = n].$$

This together with the Lipschitz continuity of $\ell$ implies the following inequality for all $z, z' \in \mathcal{Z}$

$$\left|\ell(\mathbf{w}_{t+1}; z, z') - \ell(\mathbf{w}_{t+1}'; z, z')\right| \leq L\|\mathbf{w}_{t+1} - \mathbf{w}_{t+1}'\|_2$$
$$\leq 2L^2 \sum_{k=1}^{t} \eta_k \mathbb{I}[i_k = n \text{ or } j_k = n].$$

The proof is complete. $\square$

We now apply the above uniform stability bounds and Theorem 1 to prove Theorem 4.

*Proof of Theorem 4.* We can apply Theorem 1 with $A(S) = \mathbf{w}_T$ and the uniform stability bounds in Lemma C.3 to show with probability at least $1 - \delta/2$ that

$$\left|R_S(\mathbf{w}_T) - R(\mathbf{w}_T)\right| = O\left((\log n \log(1/\delta)) \sum_{t=1}^{T} \eta \mathbb{I}[i_t = n \text{ or } j_t = n]\right) + O(n^{-\frac{1}{2}}\sqrt{\log(1/\delta)}),$$

(C.1)

where $\eta = c/\sqrt{T}$. Let $X_t = \mathbb{I}[i_t = n \text{ or } j_t = n]$. It is clear that

$$\mathbb{E}[X_t] = \Pr\{i_t = n \text{ or } j_t = n\} \leq \Pr\{i_t = n\} + \Pr\{j_t = n\} = 2/n.$$

Applying Lemma C.2 with $X_t = \mathbb{I}[i_t = n \text{ or } j_t = n]$ then gives with probability $1 - \delta/2$ that

$$\sum_{t=1}^{T} X_t \leq \left(1 + \sqrt{3\mu^{-1}\log(1/\delta)}\right)\mu,$$

where $\mu = \sum_{t=1}^{T}\mathbb{E}[X_t] \leq 2T/n$. It then follows with probability $1 - \delta/2$ that

$$\sum_{t=1}^{T} X_t \leq \frac{2T}{n}\left(1 + \sqrt{2nT^{-1}\log(1/\delta)}\right). \tag{C.2}$$

Combining (C.1) and (C.2) together, we derive the following inequality with probability $1 - \delta$

$$\left|R_S(\mathbf{w}_T) - R(\mathbf{w}_T)\right| = O(n^{-\frac{1}{2}}\sqrt{\log(1/\delta)} + T\eta(\log n\log(1/\delta))n^{-1}$$
$$+ \eta\log n\log(1/\delta)\sqrt{n^{-1}T\log(1/\delta)}).$$

The proof is complete with $\eta = c/\sqrt{T}$. $\qquad\square$

**Remark C.1.** We now give details on deriving excess risk bounds based on the estimation error bounds in Theorem 4. We can decompose the excess risk into optimization errors and estimation errors as follows (we omit $\log(1/\delta)$) [3]

$$R(\mathbf{w}_T) - R(\mathbf{w}_R^*) = R(\mathbf{w}_T) - R_S(\mathbf{w}_T) + R_S(\mathbf{w}_T) - R_S(\mathbf{w}_R^*) + R_S(\mathbf{w}_R^*) - R(\mathbf{w}_R^*)$$
$$= \left(R(\mathbf{w}_T) - R_S(\mathbf{w}_T)\right) + \left(R_S(\mathbf{w}_T) - R_S(\mathbf{w}_R^*)\right) + O(n^{-\frac{1}{2}}), \tag{C.3}$$

where we have used (B.9). The first term is the estimation error and comes from the approximation of testing errors by training errors. The second term is the optimization error which comes since the optimization algorithm may not output the exact minimizer. Then Theorem 4 actually presents estimation error bounds. If we further assume $\|\mathbf{w}_t\| \leq B$ for some $B > 0$ and all $t$, then it was shown with high probability that [9]

$$R_S(\mathbf{w}_T) - R_S(\mathbf{w}_R^*) = O(T^{-\frac{1}{2}}\log T). \tag{C.4}$$

We can plug the above optimization error bounds and the estimation error bounds in Theorem 4 into (C.3), and get with high probability

$$R(\mathbf{w}_T) - R(\mathbf{w}_R^*) = O\left(\log n\sqrt{T}/n + n^{-\frac{1}{2}}\log n\right) + O(T^{-\frac{1}{2}}\log T).$$

One can take an optimal $T \asymp n$ to trade-off the optimization and estimation errors, and get

$$R(\mathbf{w}_T) - R(\mathbf{w}_R^*) = O(n^{-\frac{1}{2}}\log n).$$

**Remark C.2.** If we plug the uniform stability bounds in Lemma C.3 into the existing connection between stability and generalization established in (4.2), we get with high probability that

$$\left|R_S(\mathbf{w}_T) - R(\mathbf{w}_T)\right| = O\left(\sqrt{n}\sum_{t=1}^{T}\eta_t\mathbb{I}[i_t = n \text{ or } j_t = n] + n^{-\frac{1}{2}}\right).$$

This together with (C.2) shows the following inequality with high probability ($\eta_t = \eta = O(1/\sqrt{T})$)

$$\left|R_S(\mathbf{w}_T) - R(\mathbf{w}_T)\right| = O\left(\frac{T\eta}{\sqrt{n}}\left(1 + \sqrt{n/T}\right) + n^{-\frac{1}{2}}\right) = O\left(\frac{\sqrt{T}}{\sqrt{n}}\left(1 + \sqrt{n/T}\right) + n^{-\frac{1}{2}}\right).$$

We can plug the above estimation error bound, the optimization error bound (C.4) back into (C.3), and derive the following excess risk bound with high probability

$$R(\mathbf{w}_T) - R(\mathbf{w}_R^*) = O\left(\frac{\log T}{\sqrt{T}} + \frac{\sqrt{T}}{\sqrt{n}}\left(1 + \sqrt{n/T}\right) + n^{-\frac{1}{2}}\right) = O(1).$$

# D Proofs on Optimistic Bounds

In this section, we prove optimistic bounds in Theorem 6 by using the smoothness of loss functions. We first prove Theorem 5 on the connection between generalization and on-average stability.

*Proof of Theorem 5.* For all $i, j \in [n]$, let $S_{i,j}$ be defined by (3.4). Due to the symmetry, we know $\mathbb{E}[R(A(S))] = \mathbb{E}[R(A(S_{i,j}))]$ for all $i, j \in [n]$ with $i \neq j$ and therefore

$$\mathbb{E}\big[R(A(S)) - R_S(A(S))\big] = \frac{1}{n(n-1)} \sum_{i,j\in[n]:i\neq j} \mathbb{E}\big[R(A(S_{i,j})) - R_S(A(S))\big]$$

$$= \frac{1}{n(n-1)} \sum_{i,j\in[n]:i\neq j} \mathbb{E}\Big[\ell\big(A(S_{i,j}); z_i, z_j\big) - \ell\big(A(S); z_i, z_j\big)\Big] \leq \gamma,$$

where the second identity holds since $A(S_{i,j})$ is independent of $z_i$ and $z_j$. The proof is complete. $\quad\square$

We then introduce some basic properties of smooth functions. For a $\alpha$-smooth and non-negative function $g$, we have the following self-bounding property [14]

$$\|g'(\mathbf{w})\|^2 \leq 2\alpha g(\mathbf{w}), \quad \forall \mathbf{w} \in \mathcal{W} \tag{D.1}$$

and the following elementary inequality

$$g(\mathbf{w}) \leq g(\mathbf{w}') + \langle g'(\mathbf{w}'), \mathbf{w} - \mathbf{w}' \rangle + \frac{\alpha\|\mathbf{w} - \mathbf{w}'\|^2}{2}, \quad \forall \mathbf{w}, \mathbf{w}' \in \mathcal{W}. \tag{D.2}$$

We then present a useful lemma.

**Lemma D.1.** *Let $S, S'$ be defined in Definition 2. Assume for all $z, z'$, $\ell(\cdot, z, z')$ is $\alpha$-smooth w.r.t. a norm. For all $i \in [n]$, let $S_i$ be defined as (A.2) and $\epsilon > 0$. Then*

$$\mathbb{E}\big[R(A(S)) - R_S(A(S))\big] \leq \frac{\alpha\mathbb{E}[R_S(A(S))]}{\epsilon} + \frac{2(\epsilon + \alpha)}{n} \sum_{i\in[n]} \mathbb{E}\big[\|A(S_i) - A(S)\|^2\big].$$

*Proof.* For all $i, j \in [n]$, let $S_{i,j}$ be defined by (3.4). According to (D.2), the Cauchy-Schwartz inequality and (D.1), for all $i, j \in [n]$ we know

$$\ell(A(S_{i,j}); z_i, z_j) - \ell(A(S); z_i; z_j) \leq \langle \ell'(A(S); z_i, z_j), A(S_{i,j}) - A(S) \rangle + \frac{\alpha}{2}\|A(S_{i,j}) - A(S)\|^2$$

$$\leq \|\ell'(A(S); z_i, z_j)\|\|A(S_{i,j}) - A(S)\| + \frac{\alpha}{2}\|A(S_{i,j}) - A(S)\|^2$$

$$\leq \frac{\|\ell'(A(S); z_i, z_j)\|^2}{2\epsilon} + \frac{\epsilon + \alpha}{2}\|A(S_{i,j}) - A(S)\|^2$$

$$\leq \frac{\alpha\ell(A(S); z_i, z_j)}{\epsilon} + \frac{\epsilon + \alpha}{2}\|A(S_{i,j}) - A(S)\|^2.$$

We can plug the above inequality into Theorem 5 to derive

$$\mathbb{E}\big[R(A(S)) - R_S(A(S))\big]$$

$$\leq \frac{\alpha}{\epsilon n(n-1)} \sum_{i\neq j} \mathbb{E}\big[\ell(A(S); z_i, z_j)\big] + \frac{\epsilon + \alpha}{2n(n-1)} \sum_{i\neq j} \mathbb{E}\big[\|A(S_{i,j}) - A(S)\|^2\big]$$

$$= \frac{\alpha\mathbb{E}[R_S(A(S))]}{\epsilon} + \frac{\epsilon + \alpha}{2n(n-1)} \sum_{i\neq j} \mathbb{E}\big[\|A(S_{i,j}) - A(S)\|^2\big]. \tag{D.3}$$

By the elementary inequality $(a + b)^2 \leq 2a^2 + 2b^2$, we get the following inequality for all $i \neq j$

$$\mathbb{E}\big[\|A(S_{i,j}) - A(S)\|^2\big] \leq 2\mathbb{E}\big[\|A(S_{i,j}) - A(S_i)\|^2\big] + 2\mathbb{E}\big[\|A(S_i) - A(S)\|^2\big]$$

$$= 2\mathbb{E}\big[\|A(S_i) - A(S)\|^2\big] + 2\mathbb{E}\big[\|A(S_j) - A(S)\|^2\big],$$

where we have used the following identity due to the symmetry between $z_i$ and $z_i'$

$$\mathbb{E}\big[\|A(S_{i,j}) - A(S_i)\|^2\big] = \mathbb{E}\big[\|A(S_j) - A(S)\|^2\big].$$

Plugging the above inequality back into (D.3), we know

$$\mathbb{E}\big[R(A(S)) - R_S(A(S))\big] \le \frac{\alpha \mathbb{E}[R_S(A(S))]}{\epsilon}$$
$$+ \frac{\epsilon + \alpha}{n(n-1)} \sum_{i \ne j} \Big( \mathbb{E}\big[\|A(S_i) - A(S)\|^2\big] + \mathbb{E}\big[\|A(S_j) - A(S)\|^2\big] \Big).$$

This yields the stated inequality and finishes the proof. $\qquad\square$

*Proof of Theorem 6.* According to Lemma B.1 and the $\alpha$-smoothness of $\ell$, we know the following inequality for any $k$

$$n(n-1)\big(F_S(A(S_k)) - F_S(A(S))\big) \le \sum_{i \in [n]:i \ne k} \Big( \Big\langle \ell'(A(S); z_i, z_k) + \ell'(A(S); z_k, z_i)$$
$$- \ell'(A(S_k); z_i, z_k') - \ell'(A(S_k); z_k', z_i); A(S_k) - A(S) \Big\rangle + \frac{4\alpha \|A(S_k) - A(S)\|^2}{2} \Big).$$

It then follows from the Cauchy-Schwartz inequality that

$$n(n-1)\big(F_S(A(S_k)) - F_S(A(S))\big) \le \sum_{i \in [n]:i \ne k} \Big( \big\|\ell'(A(S); z_i, z_k)\big\| + \big\|\ell'(A(S); z_k, z_i)\big\|$$
$$+ \big\|\ell'(A(S_k); z_i, z_k')\big\| + \big\|\ell'(A(S_k); z_k', z_i)\big\| \Big) \|A(S_k) - A(S)\| + 2\alpha(n-1)\|A(S_k) - A(S)\|^2.$$

This, together with (D.1) and (B.3), implies

$$\frac{\sigma n(n-1)\|A(S_k) - A(S)\|^2}{2} \le \sqrt{2\alpha} \sum_{i \in [n]:i \ne k} \Big( \sqrt{\ell(A(S); z_i, z_k)} + \sqrt{\ell(A(S); z_k, z_i)}$$
$$+ \sqrt{\ell(A(S_k); z_i, z_k')} + \sqrt{\ell(A(S_k); z_k', z_i)} \Big) \|A(S_k) - A(S)\| + 2\alpha(n-1)\|A(S_k) - A(S)\|^2$$

and further

$$\frac{\sigma n(n-1)\|A(S_k) - A(S)\|}{2} \le \sqrt{2\alpha} \sum_{i \in [n]:i \ne k} \Big( \sqrt{\ell(A(S); z_i, z_k)} + \sqrt{\ell(A(S); z_k, z_i)}$$
$$+ \sqrt{\ell(A(S_k); z_i, z_k')} + \sqrt{\ell(A(S_k); z_k', z_i)} \Big) + 2\alpha(n-1)\|A(S_k) - A(S)\|.$$

Since $2\alpha \le \sigma n/4$, we further get

$$\frac{\sigma n(n-1)\|A(S_k) - A(S)\|}{4} \le \sqrt{2\alpha} \sum_{i \in [n]:i \ne k} \Big( \sqrt{\ell(A(S); z_i, z_k)} + \sqrt{\ell(A(S); z_k, z_i)}$$
$$+ \sqrt{\ell(A(S_k); z_i, z_k')} + \sqrt{\ell(A(S_k); z_k', z_i)} \Big).$$

Taking a square over both sides and using the standard inequality $\big(\sum_{i=1}^{n-1} a_i\big)^2 \le (n-1)\sum_{i=1}^{n-1} a_i^2$, we derive

$$\frac{\sigma^2 n^2 (n-1)^2 \|A(S_k) - A(S)\|^2}{16} \le 2\alpha(n-1) \sum_{i \in [n]:i \ne k} \Big( \sqrt{\ell(A(S); z_i, z_k)} + \sqrt{\ell(A(S); z_k, z_i)}$$
$$+ \sqrt{\ell(A(S_k); z_i, z_k')} + \sqrt{\ell(A(S_k); z_k', z_i)} \Big)^2.$$

This, further together with the inequality $(a + b + c + d)^2 \leq 4(a^2 + b^2 + c^2 + d^2)$, implies

$$\sigma^2 n^2 (n-1) \|A(S_k) - A(S)\|^2 \leq 128\alpha \sum_{i \in [n]: i \neq k} \Big( \ell(A(S); z_i, z_k) + \ell(A(S); z_k, z_i)$$
$$+ \ell(A(S_k); z_i, z_k') + \ell(A(S_k); z_k', z_i) \Big).$$

Taking a summation of the above inequality from $k = 1$ to $n$, we get

$$\sigma^2 n^2 (n-1) \sum_{k=1}^{n} \|A(S_k) - A(S)\|^2 \leq 128\alpha \sum_{i,k \in [n]: i \neq k} \Big( \ell(A(S); z_i, z_k) + \ell(A(S); z_k, z_i)$$
$$+ \ell(A(S_k); z_i, z_k') + \ell(A(S_k); z_k', z_i) \Big). \quad \text{(D.4)}$$

Due to the symmetry, we know

$$\mathbb{E}\big[\ell(A(S_k); z_i, z_k')\big] = \mathbb{E}\big[\ell(A(S); z_i, z_k)\big], \quad \forall i \neq k.$$

It then follows that

$$\sum_{i,k \in [n]: i \neq k} \mathbb{E}\Big[\ell(A(S); z_i, z_k) + \ell(A(S); z_k, z_i) + \ell(A(S_k); z_i, z_k') + \ell(A(S_k); z_k', z_i)\Big]$$
$$= \sum_{i,k \in [n]: i \neq k} \mathbb{E}\Big[\ell(A(S); z_i, z_k) + \ell(A(S); z_k, z_i) + \ell(A(S); z_i, z_k) + \ell(A(S); z_k, z_i)\Big]$$
$$= 4n(n-1)\mathbb{E}\big[R_S(A(S))\big].$$

We can plug the above inequality back into (D.4) and derive that

$$\sigma^2 n \sum_{k=1}^{n} \mathbb{E}\big[\|A(S_k) - A(S)\|^2\big] \leq 512\alpha \mathbb{E}\big[R_S(A(S))\big].$$

We now plug the above inequality back into Lemma D.1 and derive that the following inequality for all $\epsilon > 0$

$$\mathbb{E}\big[R(A(S)) - R_S(A(S))\big] \leq \frac{\alpha \mathbb{E}[R_S(A(S))]}{\epsilon} + \frac{1024(\epsilon + \alpha)\alpha}{n^2 \sigma^2} \mathbb{E}\big[R_S(A(S))\big].$$

We can take $\epsilon = \frac{n\sigma}{32}$ to derive

$$\mathbb{E}\big[R(A(S)) - R_S(A(S))\big] \leq \Big( \frac{1024\alpha^2}{n^2 \sigma^2} + \frac{64\alpha}{n\sigma} \Big) \mathbb{E}\big[R_S(A(S))\big].$$

Furthermore, according to the definition of $A(S)$ we know ($\mathbf{w}^*$ is independent of $S$)

$$\mathbb{E}[F(A(S))] - F(\mathbf{w}^*) = \mathbb{E}\big[F(A(S)) - F_S(A(S))\big] + \mathbb{E}\big[F_S(A(S)) - F_S(\mathbf{w}^*)\big]$$
$$\leq \mathbb{E}\big[F(A(S)) - F_S(A(S))\big] = \mathbb{E}\big[R(A(S)) - R_S(A(S))\big].$$

This finishes the proof of (4.8).

We now turn to the bound of $\mathbb{E}[R(A(S))] - R(\mathbf{w}_R^*)$. Analogously to (B.8), we know

$$\mathbb{E}\big[R(A(S)) - R(\mathbf{w}_R^*)\big] \leq \mathbb{E}\big[R(A(S)) - R_S(A(S))\big] + O(\sigma \|\mathbf{w}_R^*\|^2)$$
$$= O\Big(\frac{1}{n\sigma}\Big) \mathbb{E}\big[R_S(A(S))\big] + O(\sigma \|\mathbf{w}_R^*\|^2), \quad \text{(D.5)}$$

where we have used (4.8) in the last step. According to the definition of $A(S)$, we further know

$$R_S(A(S)) + r(A(S)) \leq R_S(\mathbf{w}_R^*) + r(\mathbf{w}_R^*) = R_S(\mathbf{w}_R^*) + O(\sigma \|\mathbf{w}_R^*\|^2).$$

Since $\mathbf{w}_R^*$ is independent of $S$, we can take expectation to derive

$$\mathbb{E}[R_S(A(S))] = R(\mathbf{w}_R^*) + O(\sigma \|\mathbf{w}_R^*\|^2).$$

We can plug the above inequality back into (D.5), and derive

$$\mathbb{E}\big[R(A(S)) - R(\mathbf{w}_R^*)\big] = O\Big(\frac{R(\mathbf{w}_R^*)}{n\sigma} + O\big(n^{-1} + \sigma\big)\|\mathbf{w}_R^*\|^2\Big).$$

We can take

$$\sigma = \max\Big\{\frac{8\alpha}{n}, \sqrt{\frac{R(\mathbf{w}_R^*)}{n\|\mathbf{w}_R^*\|^2}}\Big\}$$

and derive

$$\mathbb{E}\big[R(A(S)) - R(\mathbf{w}_R^*)\big] = O\Big(\frac{\sqrt{R(\mathbf{w}_R^*)}\|\mathbf{w}_R^*\|}{\sqrt{n}} + \frac{\|\mathbf{w}_R^*\|^2}{n}\Big).$$

This establishes (4.9) and finishes the proof.

$\qquad\square$

# E   Proofs on Applications

In this section, we present proofs for applications of our general results to metric learning.

*Proof of Corollary 10.*  It is well known that $F_S$ is $2\lambda$-strongly convex w.r.t. $\|\cdot\|$. To apply Theorem 3, we require to check (4.3). For all $\mathbf{w}, \mathbf{w}', z, z'$, we know

$$
\begin{aligned}
&\big|\ell^\psi(\mathbf{w}; z, z') - \ell^\psi(\mathbf{w}'; z, z')\big| \\
&= \Big|\max\big\{0, 1 - \tau(y, y')(1 - h_\mathbf{w}(x, x'))\big\} - \max\big\{0, 1 - \tau(y, y')(1 - h_{\mathbf{w}'}(x, x'))\big\}\Big| \\
&\le |\tau(y, y')|\big|h_\mathbf{w}(x, x') - h_{\mathbf{w}'}(x, x')\big| \le \big|\langle \mathbf{w} - \mathbf{w}', (x - x')(x - x')\rangle\big| \\
&\le 4B^2\|\mathbf{w} - \mathbf{w}'\|.
\end{aligned}
$$

Therefore, (4.3) holds with $L = 4B^2$. The proof then completes by applying Theorem 3.   $\square$

*Proof of Corollary 11.*  To apply Theorem 4, it suffices to show the smoothness of the loss function. The gradient of $\ell^\psi$ w.r.t. $\mathbf{w}$ can be calculated by

$$\nabla\ell^\psi(\mathbf{w}; z, z') = -\psi'\big(\tau(y, y')(1 - h_\mathbf{w}(x, x'))\big)\tau(y, y')(x - x')(x - x')^\top.$$

Then, for any $\mathbf{w}$ and $\mathbf{w}' \in \mathcal{W}$ we have

$$
\begin{aligned}
&\big\|\nabla\ell^\psi(\mathbf{w}; z, z') - \nabla\ell^\psi(\mathbf{w}'; z, z')\big\|_K \\
&\le \big\|\tau(y, y')(x - x')(x - x')^\top\big\|\big|\psi'\big(\tau(y, y')(1 - h_\mathbf{w}(x, x'))\big) - \psi'\big(\tau(y, y')(1 - h_{\mathbf{w}'}(x, x'))\big)\big| \\
&\le 4B^2\big|\psi'\big(\tau(y, y')(1 - h_\mathbf{w}(x, x'))\big) - \psi'\big(\tau(y, y')(1 - h_{\mathbf{w}'}(x, x'))\big)\big| \\
&\le 4B^2|\tau(y, y')|\big|(1 - h_\mathbf{w}(x, x')) - (1 - h_{\mathbf{w}'}(x, x'))\big| \\
&= 4B^2\big|\langle \mathbf{w} - \mathbf{w}', (x - x')(x - x')^\top\rangle\big| \\
&\le 16B^4\|\mathbf{w} - \mathbf{w}'\|,
\end{aligned}
$$

where we have used the 1-smoothness of the logistic loss in the third step. That is, $\ell^\psi$ is $(16B^4)$-smooth w.r.t. the Frobenius norm. The stated bound then follows from Theorem 4.   $\square$

# F   Minimax Optimal Excess Risk Bounds for Pairwise Learning

Here we explain that the bound $O(n^{-\frac{1}{2}})$ is minimax optimal for the excess risks in pairwise learning. To see this, we consider pairwise loss functions which do not depend on the second example, i.e., $\ell(\mathbf{w}; z, z') = \ell(\mathbf{w}; z, \tilde{z}')$ for all $z', \tilde{z}' \in \mathcal{Z}$. Then it is clear that $R_S$ defined in (3.1) becomes

$$R_S(\mathbf{w}) = \frac{1}{n}\sum_{i\in[n]}\frac{1}{n-1}\sum_{j\in[n]:j\ne i}\ell(\mathbf{w}; z_i, z_j) = \frac{1}{n}\sum_{i\in[n]}\ell(\mathbf{w}; z_i, z_0) := \widetilde{R}_S(\mathbf{w}),$$

where $z_0$ is any fixed point in $\mathcal{Z}$. This is actually an objective function for pointwise learning. We know that for any estimator we can find a pointwise learning problem such that this estimator has the excess risk bound $O(n^{-\frac{1}{2}})$ [15]. Then, for any estimator we can build a pairwise learning problem such that this estimator has at best the excess risk bound $O(n^{-\frac{1}{2}})$. Furthermore, we can construct such a pairwise learning problem with the loss function independent of the second example.