[Reviews · NeurIPS 2020]

Review 1

Summary and Contributions: This paper gives new generalization bounds for learning problems where the loss function depends on pairs of examples. This setting comes up naturally in ranking problems and in metric learning. The approach is through uniform stability, and a natural average case variation. While uniform stability arguments have been sharpened in recent years (e.g. the work of Feldman and Vondrak) in the pointwise learning setting, there were many gaps between the lower and upper bounds in the pairwise setting. This paper manages to close many of them, in some cases improving the generalization bound by as much as a \sqrt{n} factor where n is the number of examples.

Strengths: There is a ton of work on generalization bounds, and yet this paper still manages to provide a fresh angle. First, they highlight an important class of problems, namely pairwise learning, where obtaining nearly tight bounds is considerably harder, in large part because the objective function is no longer the sum of iid random variables. Second, they give a general methodology for obtaining sharper connections between stability and generalization in this setting. If an algorithm is \gamma uniformly stable they give a generalization bound of \gamma \log n that holds with high probability, which improves upon the previously best known bound of \gamma \sqrt{n}. Third they introduce a refined notion of on-average stability that allows them to prove better bounds in optimistic cases (i.e. when the best model has small error). Fourth give compelling applications to generalization bounds for ranking and metric learning. A notable highlight is that these include the first generalization bounds for SGD in pairwise learning. Existing bounds could not obtain non-trivial guarantees.

Weaknesses: I don't think the paper has any particular weaknesses.

Correctness: I have not verified all the proofs, but the ones I have looked through are all correct.

Clarity: The paper is well-written. There are a lot of results in the paper, but nevertheless it is easy to navigate given how the results build on each other.

Relation to Prior Work: The paper does a nice job of surveying the relevant generalization literature. It also gives some nice intuitions about the relative strengths and weaknesses of different approaches (e.g. the ability of stability approaches to give dimension independent guarantees).

Reproducibility: Yes

Additional Feedback: I have read the author response. I agree with the comment pointed out by another reviewer that the results are not quite as optimal as they were claimed, however I still think this is a compelling paper with both an interesting overarching message and neat theoretical applications.


Review 2

Summary and Contributions: This paper provided a refined stability analysis by developing generalization bounds, and further apply these results to ranking and metric learning. Main contributions: This paper extended the stablity analysis of pointwise learning [4] to pairwise learning, which can be $\sqrt{n}$-times faster than the existing results.They further apply the above stablity-based results to ranking and metric learning.

Strengths: 1: Theoretical: This paper derived a refined stability analysis for pairwise learning, which is faster than the existing results. 2: They apply the refined stablity-based results for ranking, SGD-based ranking and metric learning. They give the first high-probability generalization bound for SGD in pairwise learning.

Weaknesses: The main contribution of this manuscript is the extension of the stablity analysis of pointwise learning to pairwise learning. However, the proof techniques is mainly based on [4], which limits the novelty of manuscript.

Correctness: Some claims are not rigorous. The proofs seems to be correct, I do not check the proof details lines by lines.

Clarity: This paper is well wirtten.

Relation to Prior Work: Yes

Reproducibility: Yes

Additional Feedback: In lines 208-210, the authors claims that "the bound $O(n^{-1/2}\log(n))$ proposed is mimimax optimal". This claim is not correct, because of the conditions of the upper bound is different from the low bound. To obtian the upper bound of $O(n^{-1/2}\log(n))$, the loss function should be strongly convex, but for the lower bound is not needed. If you claim a bound is mimimax optimal, the conditions of upper and lower bounds must the same. The condition of $\|w_R^\ast\|=O(1)$ may be too strict. Usually, the norm of w is dependent on the dimension of w. In this manuscript, they claims that they don't require the conditoin of the "bounded" loss function, but for most of the results the loss should be strongly convex. Minor comments: Lines 130: What's the "RRM" means? Lines 197: $\lambda$->$\sigma$


Review 3

Summary and Contributions: The paper adresses the generalization of pairwise learning (minimization of an expected loss depending on independent data pairs) under the perspective of algorithmic stability. Pairwise learning is relevant to ranking and metric learning. Theorem 1, the first bound on the generalization gap on which most other results in the paper are based, relies on uniform stability of the training algorithm, as introduced by Boulsquet/Elisseeff in 2002. The bound is of O(\gamma log n + n^{-1/2}, where \gamma is the stability parameter and n the sample size, and it also replaces the frequenttly assumed boundedness of the loss function by a uniform bound on the sample-expectation of the loss of the algorithms output. Theorem 3 then considers regularized algorithms minimizing a strongly convex objective under Lipschitz assumptions, and second-moment assumptions on the minimizer of the regularized risk. Uniform stability is verified and corresponding bounds are derived from Theorem 1. Under second-moment assumptions on the minimizer of the true risk there is also a bound on the excess risk. Theorem 4 gives a generalization bound for stochastic gradient descent. Under the condition of \alpha-smoothness of the loss function uniform stability is demonstrated, depending on the number of training iterations. Again Theorem 1 is invoked to give the result. Theorem 6 then gives a bound on the excess risk in expectation, of order \sqrt{R/n} + 1/n, where R is the risk of the true-risk-minimizer. A final section applies the above results to ranking and metric learning. -----------------Update---------------------------------------------------- I have read the authors response and the other reviews and I keep my score at 7

Strengths: The paper extends state of the art results in the theory of algorithmic stability from pointwise to pairwise learning. The bounds given are powerful and, in the context of pairwise learning, novel. The proof techniques are sophisticated, but presented with sufficient clarity.

Weaknesses: I have very little substantial criticism. Perhaps the second paragraphs of section 2 exaggerates the merits of algorithmic stability a little bit. Clearly uniform bounds can also provide very valuable information in non-convex settings when exact minimization is impossible and algorithmic stability is limited to the results in [21]. I also think reference [7] (Bousquet et al 2019) could have been acknowledged a bit more. Apart from the tricky decomposition of U-statistics, the proof of Theorem 1 is a fairly direct application of the elegant techniques of [7].

Correctness: I checked the proofs in appendices A and B. It all seemed sound and well explained, but I studied the proof of Theorem 4 (Appendix C) only superficially, and I didn't read the proof of Theorem 6.

Clarity: Well written and polished with very few typos. l116 \subseteq should replace \in l140 insenstitive to perturbations l182 arg inf should be arg min (shouldn't \cal{W} be closed if it is bounded?) l197 \lambda should be \sigma l198 shouldn't the "mild assumption" in l122 (Appendix B) be stated here? l226 "and any" should be "and for any"

Relation to Prior Work: Generally quite well, but see my remark above w.r.t. reference [7]

Reproducibility: Yes

Additional Feedback:

[Author Response · NeurIPS 2020]

**Reviewer #3**

Thank you for your careful reading and very positive comments. We feel very encouraged.

**Reviewer #4**

Thank you for your careful reading and very constructive comments.

**Q**: *Proof techniques are mainly based on [4] (Bousquet et al. 2019) which limits the novelty.*

**A**: Thank you for your valuable review. We would like to emphasize the novelty of our work. As you point out, one of
the key tools we make use of is a concentration inequality from [4], which considers a summation of $n$ functions of $n$
independent random variables. However, this concentration inequality does not fit the structure of pairwise learning. To
address this discrepancy, we introduce novel and tricky decompositions of U-statistics (cf. Lines 33-34 and Lines 38-39
in the Appendix). Furthermore, the existing stability analysis (presented in [4]) requires the assumption of a bounded
loss: for RRM (Regularized Risk Minimization), the loss needs to be bounded by $O(1/\sqrt{\sigma})$ (Lines 185-187). Thus,
even in the case of *pointwise learning*, the stability analysis in [4] only yields loose bounds of rate $O((n\sigma)^{-1/2})$ when
one takes into account the magnitude of the loss function—in contrast, we prove $O((n\sigma)^{-1} + n^{-1/2})$ in that case (cf.
eq. (4.4)). This boundedness assumption can not be addressed by adding a constraint, as doing so would negate the
main advantage of the RRM approach. We relax this boundedness assumption to a variance assumption on $\ell(\mathbf{w}^*; Z, \tilde{Z})$.
Note that the expectation of $\ell(\mathbf{w}^*; Z, \tilde{Z})$ is $R(\mathbf{w}^*)$, which is small according to the definition of $\mathbf{w}^*$. Therefore, it is
reasonable to assume that the variance of $\ell(\mathbf{w}^*; Z, \tilde{Z})$ is bounded. To achieve this relaxation, we use a novel application
of Theorem 1 to $\tilde{\ell}(\mathbf{w}; z, \tilde{z}) = \ell(\mathbf{w}; z, \tilde{z}) - \ell(\mathbf{w}^*; z, \tilde{z})$ instead of $\ell(\mathbf{w}; z, \tilde{z})$ (cf. Line 107 in the Appendix). Moreover,
we introduce a novel lemma (Lemma 2) to show $\left|\mathbb{E}_S\left[\tilde{\ell}(A(S); z, \tilde{z})\right]\right| = O(1/(\sqrt{n}\sigma))$ (cf. Line 105 in the Appendix)
and apply Bernstein's inequality to address the concentration behavior of $\ell(\mathbf{w}^*; Z, \tilde{Z})$. We believe the techniques we
develop, both to adapt to the structure of pairwise learning and to relax the boundedness assumption, are novel.

Other than this contribution, we would like to emphasize that we derive the first generalization bounds for SGD in
pairwise learning, a setting where the existing bounds could not obtain non-trivial guarantees. We also introduce
on-average stability for pairwise learning and use it to develop optimistic bounds in a low noise setting.

**Q**: *The claim on minimax optimality is not correct since the conditions do not match in upper and lower bounds.*

**A**: Thank you for your insightful comment. We promise to tone down our original informal formulation of the claim to
reflect the facts about the conditions. We would like to mention that we also obtain the bound $O(n^{-\frac{1}{2}} \log n)$ for SGD,
which does not require the objective function to be strongly convex. Thus, our bound for SGD is minimax optimal up to
a logarithmic factor. We will study lower bounds of pairwise learning in the strongly convex setting in our future study.

**Q**: *The condition $\|\mathbf{w}_R^*\| = O(1)$ may be too strict.*

**A**: Thank you for indicating this. We use this condition to simplify the presentation of eq. (4.9). If we do not impose this
condition, the upper bound in (4.9) becomes $O\left(\sqrt{R(\mathbf{w}_R^*)}\|\mathbf{w}_R^*\| n^{-\frac{1}{2}} + \|\mathbf{w}_R^*\|^2 n^{-1}\right)$ (cf. Line 227 in the Appendix).

**Q**: *They don't require the condition of bounded loss function, but for most of results the loss should be strongly convex.*

**A**: Thank you. To be precise, we require the strong convexity of $F_S$ (loss function + regularizer), not that of loss
functions. Previous work assumed loss functions to be bounded by a universal constant, as well as the strong convexity
of $F_S$. However, strong convexity only guarantees that loss functions are bounded by $O(\sqrt{1/\sigma})$ (Line 185-187). Thus
without a boundedness assumption, the existing stability analysis implies a loose bound even for *pointwise learning* [4].

**Q**: *Meaning of RRM in Line 130. $\lambda$ should be $\sigma$ in Line 197.* **A**: Thank you. RRM stands for "regularized risk
minimization". We explain this abbreviation in Line 44. We will replace $\lambda$ by $\sigma$ in Line 197.

**Reviewer #6**

Thank you for your careful reading and very positive comments. We feel very encouraged.

**Q**: *The second paragraph of Section 2 exaggerates the merits of algorithmic stability a little bit.*

**A**: We agree with this point. We will further discuss the merit of the uniform convergence approach over the stability
approach in providing very valuable information in the non-convex setting.

**Q**: *Reference [7] (Bousquet et al. 2019) could have been acknowledged a bit more.*

**A**: We agree and promise to acknowledge this important reference more.

**Q**: *Some Typos.* **A**: Thank you for your careful reading. We completely agree and will fix them in the revision.

[Meta-Review · NeurIPS 2020]

This paper presents a generalization bound for pairwise learning. Overall the reviewers found it interesting, novel and the proof techinique sophisticated. Please make sure to address Reviewer #4 comment on the minimax optimality, as you ackgnoweledged in your response.